# MEDSIGHT: Towards Grounded Visual Comprehension in Medical Large Vision-Language Models

Aofei Chang[* 1]   Le Huang[2]   Alex James Boyd[2]   Parminder Bhatia[2]   Taha Kass-Hout[2]
Fenglong Ma[1]   Cao Xiao[2]

## Abstract

Medical large vision-language models (Med-LVLMs) have recently achieved remarkable progress in vision–language comprehension and medical image segmentation. However, existing models still struggle to unify these two capabilities, which is essential for achieving clinically reasoning that connects visual findings with semantic interpretation. We present **MEDSIGHT, a unified framework that equips Med-LVLMs with structured, pixel-level understanding for grounded visual comprehension**. MEDSIGHT introduces a novel *Region Perceiver* module that produces region-centric tokens, encoding spatial information directly into representation space of the language model. We further propose a medical region codebook into the LLM vocabulary, allowing the model to generate discrete region codes as symbolic representations of anatomical and pathological regions. These codes are decoded through the Region Perceiver to reconstruct segmentation masks, achieving end-to-end spatial grounding. Lastly, MEDSIGHT combines Region Perceiver, Codebook and LLM using our proposed progressive training strategy to gradually aligns these modules stably. Trained on only 72K multimodal instruction pairs, MedSIGHT achieves state-of-the-art performance across diverse imaging modalities on both medical comprehension and segmentation tasks. Code and model are publicly available at GitHub.

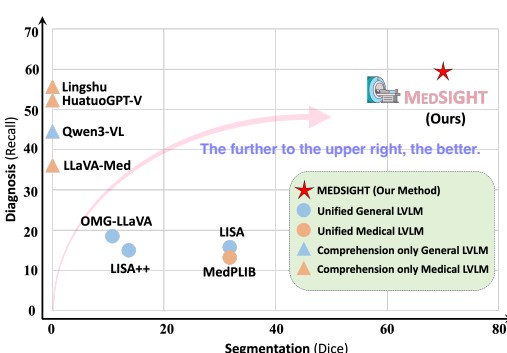

*Figure 1.* The performance comparison of Med-LVLMs on Grounded Diagnostic Segmentation.

## 1. Introduction

Recent advances in Med-LVLMs (Li et al., 2023a; Chen et al., 2024a; Lin et al., 2025) have led to remarkable progress in the multimodal understanding of medical data. However, most existing methods remain limited to high-level visual comprehension and lack the ability to ground their responses at the pixel level, which is crucial for precise localization of pathological regions and for ensuring that model reasoning aligns with clinically verifiable visual evidence. To enhance medical visual grounding, recent studies have incorporated external segmentation modules into Med-LVLMs. For instance, following LISA (Lai et al., 2024), MedPLIB (Huang et al., 2025) integrates a large segmentation model SAM-Med2D (Cheng et al., 2023) and instructs the LLM to output a special token (e.g., `[SEG]`) to trigger region segmentation. Despite providing vision grounding, such a design still faces the following two key limitations.

First, existing Med-LVLMs rely solely on *patch-level* visual features extracted from CLIP-based encoders (Radford et al., 2021) as their visual **input**. This architecture is inherently inadequate for fine-grained medical understanding because patch tokens from the upper layers of CLIP primarily encode high-level semantic context while discarding crucial spatial details, such as lesion boundaries, organ contours, and tissue textures. Consequently, these models often fail to generate diagnostic reasoning or segmentation outputs that faithfully align with the underlying visual evidence

---

[*]Work done during an internship at GE HealthCare. [1]College of Information Sciences and Technology, The Pennsylvania State University, State College, PA, USA [2]GE Healthcare, Bellevue, WA, USA. Correspondence to: Le Huang <Lena.Huang@gehealthcare.com>, Fenglong Ma <fenglong@psu.edu>, Cao Xiao <Cao.Xiao@gehealthcare.com>.

*Proceedings of the 43rd International Conference on Machine Learning*, Seoul, South Korea. PMLR 306, 2026. Copyright 2026 by the author(s).

*Table 1.* **Comparison of Med-LVLMs by model type, input granularity, and output capabilities.** Models vary substantially in how they process visual inputs (patch/region) and what outputs they produce (diagnosis, segmentation, or region tokens). **MEDSIGHT** is the only unified model that supports fine-grained perception via pixel- and region-level information in region-level inputs, while generating region tokens and pixel-level masks within a single generative framework. "Comp.-Only" denotes models trained solely on the visual comprehension task, whereas "Unified" models are trained on both visual comprehension and segmentation tasks.

| Model | Model Type | Input Granularity | | Output Capabilities | | |
|---|---|---|---|---|---|---|
| | | Patch | Region | Region Token | Segmentation | Diagnosis |
| LLaVA-Med (Li et al., 2023a) | Comp.-Only | ✓ | ✗ | ✗ | ✗ | ✓ |
| MedFlamingo (Moor et al., 2023) | Comp.-Only | ✓ | ✗ | ✗ | ✗ | ✓ |
| HuatuoGPT-Vision (Chen et al., 2024a) | Comp.-Only | ✓ | ✗ | ✗ | ✗ | ✓ |
| HealthGPT (Lin et al., 2025) | Comp.-Only | ✓ | ✗ | ✗ | ✗ | ✓ |
| Lingshu (Xu et al., 2025) | Comp.-Only | ✓ | ✗ | ✗ | ✗ | ✓ |
| VividMed (Luo et al., 2024) | Unified | ✓ | ✗ | ✗ | ✓ | ✓ |
| MedPLIB (Huang et al., 2025) | Unified | ✓ | ✗ | ✗ | ✓ | ✓ |
| **MEDSIGHT** | Unified | ✓ | ✓ | ✓ | ✓ | ✓ |

(Figure 1). Second, on the **output** side, representing all segmentation regions with only a *single token* significantly restricts the LLM's expressive capacity. This simplistic representation prevents the model from distinguishing among different anatomical structures or pathological findings and limits its ability to produce diverse, region-specific outputs necessary for accurate visual grounding.

**Motivation.** A truly grounded Med-LVLM should both perceive fine-grained visual details and express diverse, region-level understanding through language. Achieving this goal requires enhancing both the model's inputs and outputs.

**Our Solution.** To achieve these goals, we propose MED-SIGHT, a unified medical LVLM that enables grounded visual comprehension within a single generative framework, as shown in Figure 2. MEDSIGHT introduces two complementary modules that jointly enhance the model's perception and expression capabilities.

**(1) Region Perceiver for Fine-grained Perception.** To enrich the visual input to the LLM, we design a *Region Perceiver* that bridges the gap between patch-level features and pixel-level understanding. Unlike standard CLIP encoders and the Q-Former-like (Li et al., 2023b) perceiver that rely on fixed-scale image features, the Region Perceiver progressively upsamples visual features and refines multi-scale features through a *dual cross-attention* between image features and a set of learnable region queries. This design allows each region token to capture both global semantic context and fine-grained detail, effectively serving as visual anchors for reasoning. By compressing pixel-level information into a compact set of region embeddings, the Region Perceiver achieves fine-grained perception while maintaining efficiency with minimal input tokens to the LLM.

**(2) Modality-aware Region Codebook for Discrete Expression.** While the Region Perceiver enhances visual perception, the LLM must also express this visual understanding through language tokens. Existing LVLMs typically achieve visual grounding using a single special token to

denote segmentation regions, which severely limits their ability to represent diverse anatomical and pathological structures. To overcome this limitation, we introduce a medical *region codebook* that discretizes the continuous region embeddings from the Region Perceiver into interpretable code tokens. Each imaging modality (e.g., CT, MRI, etc.) is assigned a dedicated set of discrete region codes capturing modality-specific anatomical semantics. Pretrained via vector quantization, these codes are incorporated into the LLM vocabulary as learnable embeddings, enabling it to generate multiple, semantically grounded region tokens corresponding to distinct clinical concepts and anatomical entities.

Integrating the Region Perceiver, region codebook, and LLM is *non-trivial*, as these components operate in distinct representational spaces with potential dimensional mismatches and semantic misalignment. To address this, we develop **a progressive training pipeline** that gradually aligns these modules, ensuring stable optimization and coherent cross-modal integration. As illustrated in Algorithm 1, we first pre-train the Region Perceiver on medical datasets to learn spatially grounded region embeddings, aligning them with the LLM space via a vision-to-text projector. Next, we construct a modality-aware region codebook and incorporate it into the LLM vocabulary, while a text-to-vision projector maintains consistent mapping between generated region tokens and their pixel-level grounding. Finally, we jointly fine-tune the all modules through grounded instruction tuning, enabling MEDSIGHT to perform reasoning, localization, and pixel-level segmentation in a unified end-to-end generative framework.

**Contributions.** This work makes the following key contributions: (1) We propose **MEDSIGHT**, a medical large vision–language model (Med-LVLM) that unifies visual comprehension, grounding, and segmentation within a single generative framework. (2) We develop a *Region Perceiver* for fine-grained perception and a *modality-aware region codebook* for discrete, interpretable visual expression, to-

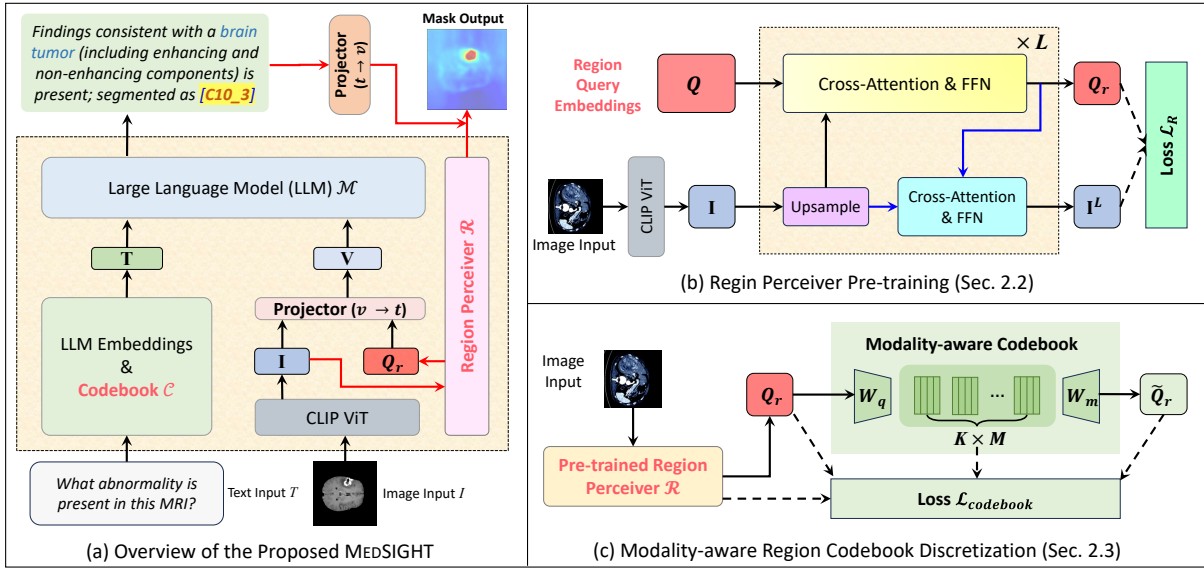

*Figure 2.* Overall framework of the proposed method. (a) Overview of our end-to-end generative framework MEDSIGHT, which supports both fine-grained visual perception and pixel-level grounding in its outputs. (b) Architecture of the Region Perceiver and its pre-training process. (c) Design of the modality-aware region codebook, which discretizes region embeddings from the pretrained Region Perceiver.

gether with a progressive multi-stage training pipeline that integrates them into the LLM for stable and coherent multimodal learning. (3) We introduce a new benchmark, **DiagSeg**, for grounded diagnostic segmentation, enabling joint evaluation of both visual diagnostic reasoning and spatial grounding. (4) We demonstrate through extensive experiments across diverse medical benchmarks that MEDSIGHT achieves state-of-the-art and interpretable performance in both visual comprehension and segmentation.

## 2. Methodology

### 2.1. Overview

Figure 2 presents an overview of the proposed model, MED-SIGHT, which takes a medical image $I$ and a text prompt $T$ as input. The image $I$ is first encoded into a patch-level embedding $\mathbf{I}$ via a pre-trained image encoder $\mathcal{E}$. This embedding is then processed by the designed region perceiver $\mathcal{R}$ (see §Sec. 2.2) to produce region-level embeddings $\mathbf{Q}_r$. The concatenation of $\mathbf{I}$ and $\mathbf{Q}_r$ is subsequently mapped into a shared multimodal space through a projection layer $\mathcal{P}_{v \to t}$, yielding the projected visual embeddings $\mathbf{V}$. Meanwhile, the vocabulary of the LLM is expanded by appending a modality-aware region codebook $\mathcal{C}$, learned in §Sec. 2.3. The learnable codebook embeddings $\mathbf{C}$, the token embeddings $\mathbf{T}$ of the text $T$, and the projected visual embeddings $\mathbf{V}$ are jointly fed into the LLM for multimodal reasoning. This design enables the LLM to generate both textual responses containing region codes and pixel-level masks, which are decoded by the pre-trained region perceiver $\mathcal{R}$. Overall, this unified architecture facilitates interpretable

region grounding and visual reasoning in an end-to-end manner, as shown in Algorithm 1. Next, we elaborate on the two key components of MEDSIGHT: (1) the pre-training and alignment of the region perceiver $\mathcal{R}$ and the modality-aware region codebook $\mathcal{C}$, and (2) the end-to-end fine-tuning of the entire framework using an instructional dataset for unifying reasoning and grounding.

### 2.2. Region Perceiver

To capture fine-grained spatial details beyond patch-level features, we present Region Perceiver, which introduce a set of learnable region query tokens $\mathbf{Q} \in \mathbb{R}^{N \times d}$ that serve as adaptive anchors encoding both semantic and spatial knowledge, where $N$ denotes the number of possible regions and $d$ is the latent dimensionality.

As illustrated in Figure 2(b), the Region Perceiver $\mathcal{R}$ is composed of $L$ iterative layers. In each layer $l$, $\mathcal{R}$ takes as input the region query tokens $\mathbf{Q}^{l-1}$ and the image embedding $\mathbf{I}^{l-1}$, where $\mathbf{Q}^0$ represents the initialized region queries and $\mathbf{I}^0$ corresponds to the output of the image encoder $\mathcal{E}$, i.e., $\mathbf{I}$. To enhance spatial granularity, we first upsample the low-resolution patch-level embedding $\mathbf{I}^{l-1}$ using a lightweight convolutional adapter, yielding finer visual embeddings $\mathbf{E}^{l-1} = \text{ConvAdapter}(\mathbf{I}^{l-1})$, following (Cheng et al., 2021).

We then introduce a **dual cross-attention** mechanism that enables mutual refinement between the fine-grained visual feature maps and the region query tokens. Each cross-attention block is followed by a feed-forward network (FFN) to further refine the updated representations. Specifically,

**Algorithm 1** Overall Training Pipeline of MEDSIGHT

1: **Input:**
- $\mathcal{D}_{\text{seg}}$: Segmentation / detection dataset;
- $\mathcal{D}_{v \to t}$: Vision→text alignment dataset;
- $\mathcal{D}_{t \to v}$: Text→vision grounding dataset;
- $\mathcal{D}_{\text{inst}}^r, \mathcal{D}_{\text{inst}}^g$: Instruction-tuning datasets;

2: **Initialize:**
- Pretrained: image encoder $\mathcal{E}$, LLM $\mathcal{M}$;
- Randomly initialized: Region Perceiver $\mathcal{R}$, Codebook $\mathcal{C}$, projectors $(\mathcal{P}_{v \to t}, \mathcal{P}_{t \to v})$;

3: **Visual Representation Preparation** (§Sec. 2.2)
- Train Region Perceiver $\mathcal{R}$ on $\mathcal{D}_{\text{seg}}$;
- Align visual features with language space using $\mathcal{P}_{v \to t}$ on $\mathcal{D}_{v \to t}$;

4: **Codebook Learning and Alignment** (§Sec. 2.3)
- Learn modality-aware region codebook $\mathcal{C}$ on $\mathcal{D}_{\text{seg}}$;
- Integrate region tokens into LLM vocabulary as $\mathbf{C}$;
- Align text-to-vision grounding via $\mathcal{P}_{t \to v}$ on $\mathcal{D}_{t \to v}$;

5: **Unified Grounded Instruction Tuning** (§Sec. 2.4)
- Jointly fine-tune $(\mathcal{M}, \mathcal{P}_{v \to t}, \mathcal{P}_{t \to v}, \mathbf{C})$ with $\mathcal{D}_{\text{inst}}^r \cup \mathcal{D}_{\text{inst}}^g$;

6: **Output:**
- Final MEDSIGHT with region-aware reasoning, visual grounding, and pixel-level segmentation.

---

the *region-to-image* attention treats $\mathbf{Q}^{l-1}$ as queries and the upsampled image features $\mathbf{E}^{l-1}$ as keys and values, allowing region queries to aggregate fine-grained spatial information via

$$\mathbf{Q}^l = \text{FFN}_r\left(\text{CrossAtt}_{r \to i}(\mathbf{Q}^{l-1}, \mathbf{E}^{l-1})\right).$$

The refined queries $\mathbf{Q}^l$ then guide the *image-to-region* attention, where $\mathbf{E}^{l-1}$ serves as the query and $\mathbf{Q}^l$ as the key and value, enabling the image representation to be updated under region-level semantic supervision:

$$\mathbf{I}^l = \text{FFN}_i\left(\text{CrossAtt}_{i \to r}(\mathbf{E}^{l-1}, \mathbf{Q}^l)\right).$$

This iterative feedback mechanism progressively enriches both spatial and contextual representations across layers.

After $L$ refinement stages, the Region Perceiver $\mathcal{R}$ produces the final region embeddings $\mathbf{Q}_r = \mathbf{Q}^L \in \mathbb{R}^{N \times d}$ and high-resolution image features $\mathbf{I}_r = \mathbf{I}^L$. These outputs are utilized for downstream segmentation and classification tasks via two lightweight supervision heads: a segmentation head that predicts region masks and a classification head that predicts region categories.

**Region Perceiver Pre-training.** We pre-train Region Perceiver on medical segmentation and detection datasets $\mathcal{D}_{\text{seg}}$ curated by BiomedParse (Zhao et al., 2025). Following standard DETR-style segmentation frameworks (Carion et al.,

2020; Cheng et al., 2021), the segmentation supervision $\mathcal{L}_{\text{seg}}$ combines Binary Cross-Entropy (BCE) and Dice losses, while the classification head is trained using cross-entropy loss $\mathcal{L}_{\text{ce}}$.[1] Hungarian matching is used to establish a one-to-one correspondence between predicted and ground-truth regions. The overall pre-training objective is defined as:

$$\mathcal{L}_{\mathcal{R}}(\mathbf{I}_r, \mathbf{Q}_r) = \mathcal{L}_{\text{seg}}(\mathbf{I}_r, \mathbf{Q}_r) + \mathcal{L}_{\text{ce}}(\mathbf{Q}_r), \qquad (1)$$

where $\mathcal{L}_{\text{seg}}(\mathbf{I}_r, \mathbf{Q}_r) = \lambda_1 \mathcal{L}_{\text{BCE}}(\mathbf{I}_r, \mathbf{Q}_r) + \lambda_2 \mathcal{L}_{\text{Dice}}(\mathbf{I}_r, \mathbf{Q}_r)$. This pre-training stage equips $\mathcal{R}$ with fine-grained, semantically aligned region embeddings, serving as a robust perceptual foundation for downstream multimodal alignment.

**Vision-to-Text Alignment.** To ensure that the patch embeddings $\mathbf{I}$ and region embeddings $\mathbf{Q}_r$ are semantically aligned with the language space, we first concatenate them and map the concatenation $[\mathbf{I}; \mathbf{Q}_r]$ to the language space through the vision-to-text projector $\mathcal{P}_{v \to t}$, yielding $\mathbf{V} = \mathcal{P}_{v \to t}([\mathbf{I}; \mathbf{Q}_r])$. Specifically, the projected visual embeddings $\mathbf{V}$ and the text embeddings $\mathbf{T}$ are fed into the LLM $\mathcal{M}$, and the standard language modeling loss $\mathcal{L}_{\text{LLM}}$ is optimized on the vision-language alignment dataset $\mathcal{D}_{v \to t}$. During this stage, we freeze the parameters of the LLM $\mathcal{M}$, the CLIP-based image encoder $\mathcal{E}$, and the pre-trained Region Perceiver $\mathcal{R}$, updating only the projector $\mathcal{P}_{v \to t}$ for alignment. It is important to note that the region codebook $\mathcal{C}$, introduced in the following subsection, is not used in this alignment stage.

### 2.3. Modality-aware Region Codebook

Although the learned region embedding $\mathbf{Q}_r$ from the Region Perceiver enables the model to achieve fine-grained understanding from visual inputs, a fundamental question remains: *"Can the LLM also express region-level understanding through language?"* To achieve this, we introduce a codebook that discretizes region embedding $\mathbf{Q}_r$ into discrete yet interpretable visual concepts. To ensure stable and consistent quantization across diverse medical imaging modalities, we design a **modality-aware codebook**:

$$\hat{\mathbf{C}} = \{\mathbf{c}_{k,m}\}_{k=1,m=1}^{K,M}, \quad \mathbf{c}_{k,m} \in \mathbb{R}^{d_c},$$

where each imaging modality $k$ maintains $M$ discrete region codes. As shown in Figure 1(b), for each region embedding $\mathbf{Q}_r^i \in \mathbb{R}^d$, we project it to the quantization space via $\mathbf{W}_q \in \mathbb{R}^{d_c \times d}$ and assign it to the nearest code:

$$\hat{\mathbf{Q}}_r^i = \mathbf{c}_{k^*, m^*} \text{ and } \{k^*, m^*\} = \arg\min_{k,m} \left\| \mathbf{W}_q \mathbf{Q}_r^i - \mathbf{c}_{k,m} \right\|_2^2.$$

The quantized region embeddings $\hat{\mathbf{Q}}_r$ are then mapped back to the vision space using $\mathbf{W}_m \in \mathbb{R}^{d \times d_c}$:

$$\tilde{\mathbf{Q}}_r = \mathbf{W}_m \hat{\mathbf{Q}}_r.$$

---

[1]Additional details are provided in the Appendix A.1.

**Codebook Pre-training.** The codebook is optimized using the standard vector quantization loss $\mathcal{L}_{\text{VQ}}$ (Van Den Oord et al., 2017; Razavi et al., 2019) and an $\ell_2$ reconstruction loss $\mathcal{L}_{\text{recon}} = \|\tilde{\mathbf{Q}}_r - \mathbf{Q}_r\|_2^2$. To maximally preserve the spatial grounding encoded in the region embeddings, we further include the loss constraint $\mathcal{L}_{\mathcal{R}}$ (i.e., Eq. (1)) used in region perceiver pre-training, forming the final objective[2]:

$$\mathcal{L}_{\text{codebook}} = \mathcal{L}_{\text{VQ}} + \mathcal{L}_{\text{recon}} + \mathcal{L}_{\mathcal{R}}(\mathbf{I}_r, \tilde{\mathbf{Q}}_r).$$

Building on the pretrained Region Perceiver, the codebook is trained on the same medical segmentation and detection datasets $\mathcal{D}_{\text{seg}}$ to ensure consistent spatial grounding. During this stage, only the codebook, $\mathbf{W}_q$ and $\mathbf{W}_m$ are updated, while the Region Perceiver remains frozen.

**Codebook and LLM Vocabulary Integration.** Let $\mathcal{C} = \{\mathbf{c}_{k,m}\}_{k=1,m=1}^{K,M}$ be the discrete region representations quantified from the continuous codebook representation $\hat{\mathbf{C}}$. We then append $\mathcal{C}$ to the LLM's vocabulary, allowing the LLM to recognize and generate region concepts as explicit textual tokens. To initialize the embeddings of these new region tokens, the codebook vectors are first projected back into the visual feature space using the trained $\mathbf{W}_m$, forming $\mathbf{C}_v \in \mathbb{R}^{(KM) \times d}$. Since $\mathbf{C}_v$ lies in the same space as the region embeddings $\mathbf{Q}_r$, we employ the trained vision-to-text projector $\mathcal{P}_{v \to t}$ to map them into the LLM's embedding space, obtaining the initialized token embeddings: $\mathbf{C} = \mathcal{P}_{v \to t}(\mathbf{C}_v)$.

**Text-to-Vision Alignment.** One of the key advantages of the proposed MEDSIGHT framework is its ability to perform region grounding and visual reasoning simultaneously, without the need for an additional segmentation module. MEDSIGHT achieves this by treating the generated region codes as triggers within the LLM's responses. When the LLM produces a region token, its hidden states $\mathbf{H}_t$ in the LLM embedding space, which encode the spatial and semantic intent of that region, serve as cues to initiate pixel-level segmentation.

To ensure that the LLM effectively comprehends and aligns with the pixel-level semantics represented by the newly introduced region embeddings $\mathbf{C}$, we introduce a text-to-vision alignment stage. Specifically, we employ a text-to-vision projector $\mathcal{P}_{t \to v}$ to map $\mathbf{H}_t$ back into the visual feature space of the Region Perceiver:

$$\mathbf{H}_v = \mathcal{P}_{t \to v}(\mathbf{H}_t).$$

The projected features $\mathbf{H}_v$ are then decoded by the segmentation head of the Region Perceiver to reconstruct region masks, supervised by the segmentation loss $\mathcal{L}_{\text{seg}}(\mathbf{I}_r, \mathbf{H}_v)$. This process grounds the LLM-generated region tokens in the visual space, enabling precise pixel-level segmentation that bridges linguistic reasoning with visual understanding.

During this stage, only $\mathcal{P}_{t \to v}$ and the new LLM embeddings $\mathbf{C}$ are updated, while the LLM $\mathcal{M}$, image encoder $\mathcal{E}$, vision-to-text projector $\mathcal{P}_{v \to t}$, and Region Perceiver $\mathcal{R}$ remain frozen. The overall objective jointly optimizes the language modeling loss $\mathcal{L}_{\text{LLM}}$ and the segmentation loss $\mathcal{L}_{\text{seg}}$ on the constructed text-to-vision alignment dataset $\mathcal{D}_{t \to v}$.

### 2.4. Unified Grounded Instruction Tuning

Although the previous pre-training and alignment stages ensure accurate pixel-level grounding and semantic mapping, the model has not yet learned how to jointly leverage these modules for multimodal comprehension and natural language generation. To bridge this gap, we perform a unified grounded instruction-tuning stage that enables MEDSIGHT to integrate linguistic reasoning, spatial understanding, and region-level grounding coherently.

Specifically, we unfreeze the LLM $\mathcal{M}$ and jointly fine-tune it with the region codebook $\mathbf{C}$, the projectors $\mathcal{P}_{v \to t}$ and $\mathcal{P}_{t \to v}$ in an end-to-end manner, while keeping the pretrained region perceiver $\mathcal{R}$ and image encoder $\mathcal{E}$ frozen. During training, the model learns to generate both descriptive and grounded responses. When a prompt involves spatial reasoning or segmentation, the LLM produces region tokens from the expanded vocabulary, whose hidden states are projected through $\mathcal{P}_{t \to v}$ and decoded into segmentation masks. This process forms a closed reasoning–grounding loop, enabling MEDSIGHT to **describe**, **locate**, and **segment** medical entities through language interactions. The overall optimization objective includes both the language modeling loss and the segmentation loss:

$$\mathcal{L}_{\text{final}} = \mathcal{L}_{\text{LLM}} + \mathcal{L}_{\text{seg}},$$

where $\mathcal{L}_{\text{LLM}}$ supervises language generation, and $\mathcal{L}_{\text{seg}}$ enforces spatial consistency between generated region tokens and visual grounding information.

We curate two complementary datasets for this stage: (i) a standard medical multimodal instruction-tuning dataset $\mathcal{D}_{\text{inst}}^r$ for enhancing general instruction-following and medical dialogue capabilities; and (ii) a grounding instruction tuning dataset $\mathcal{D}_{\text{inst}}^g$, in which model outputs include region code tokens (e.g., `[C2_16]`) that correspond to specific anatomical or pathological regions.

## 3. Grounded Diagnostic Segmentation

Traditional medical VQA datasets that include segmentation (e.g., MeCoVQA-Grounding (Huang et al., 2025), Biomed-Parse (Zhao et al., 2025)) often simplify the segmentation problem by explicitly specifying the segmentation target in the question prompt (e.g., "Segment the liver cancer."). While useful for directly evaluating localization accuracy, such settings overlook the diagnostic reasoning process in

---

[2]Additional loss details are provided in the Appendix A.2.

*Table 2.* Comparison of MEDSIGHT with other LVLMs and unified LVLMs on medical *visual comprehension* tasks. **Bold** and underlined text indicates the best performance and second-best performance, respectively. Comp. indicates "Comprehension". # **Params.** indicates the number of parameters of the base LLM, and # **Data** denotes the amount of data used during the LLM module fine-tuning stage. *Results in gray denote models that were partially trained on the evaluation benchmarks, while those in standard text represent zero-shot inference.* **\*MIMO** is evaluated only on the three public VQA datasets reported in the original paper, as neither the model nor the proposed dataset has been released publicly. Note that MedPLIB contains 14B parameters, with 7B active at inference under its MoE design.

| Type | Model | # Params | # Data | VQA-RAD ↑ close | VQA-RAD ↑ all | SLAKE ↑ close | SLAKE ↑ all | PathVQA ↑ close | PathVQA ↑ all | MMMU -Med ↑ | OMVQA↑ | DiagSeg- Diagnosis ↑ | Avg. ↑ |
|---|---|---|---|---|---|---|---|---|---|---|---|---|---|
| **Comp. only** | BLIP-2 | 6.7B | - | 43.4 | 36.8 | 41.6 | 35.3 | 48.5 | 28.8 | 27.3 | 26.9 | 22.3 | 34.5 |
| | LLaVA-v1.5 | 7B | 158K | 51.8 | 42.8 | 37.1 | 37.7 | 53.5 | 31.4 | 32.7 | 44.7 | 45.3 | 41.9 |
| | InstructBLIP | 7B | 364K | 61.0 | 44.8 | 66.8 | 43.3 | 56.0 | 32.3 | 25.3 | 29.0 | 35.9 | 43.8 |
| | Yi-VL | 6B | 10K | 52.6 | 42.1 | 52.4 | 38.4 | 54.9 | 30.9 | 38.0 | 50.2 | 33.0 | 43.6 |
| | InternVL2 | 8B | 7.3M | 64.9 | 49.0 | 66.6 | 50.1 | 60.0 | 31.9 | 43.3 | 54.5 | 42.3 | 51.4 |
| | Llama-3.2 | 11B | - | 68.9 | 45.5 | 72.4 | 52.1 | 62.8 | 33.6 | 39.3 | 63.2 | 46.4 | 53.8 |
| | Med-Flamingo | 8.3B | 1.3M | 58.6 | 43.0 | 47.0 | 25.5 | 61.9 | 31.3 | 28.7 | 34.9 | 34.6 | 40.6 |
| | LLaVA-Med | 7B | 60K | 60.2 | 48.1 | 58.4 | 44.8 | 62.3 | 35.7 | 30.0 | 41.3 | 35.3 | 46.2 |
| | HuatuoGPT-Vision | 7B | 647K | 69.7 | 60.0 | 69.0 | 60.1 | 63.6 | 41.3 | 43.3 | 66.4 | 51.1 | 58.3 |
| | Qwen3-VL | 8B | - | 72.8 | 53.4 | 76.9 | 66.1 | 66.9 | 36.9 | 46.7 | 75.3 | 44.5 | 59.9 |
| | InternVL3.5 | 8B | 16.3M | 67.7 | 48.9 | 82.1 | 72.2 | 67.0 | 38.6 | 46.0 | 84.5 | 44.6 | 61.3 |
| | HealthGPT-M3 | 3.8B | 1.5M | 73.7 | 55.9 | 74.6 | 56.4 | 78.7 | 39.7 | 43.3 | 68.5 | 44.1 | 59.4 |
| | HealthGPT-L14 | 14B | 1.5M | 77.7 | 58.3 | 76.4 | 64.5 | 85.9 | 44.4 | 49.2 | 74.4 | 51.2 | 64.7 |
| | Lingshu-7B | 7B | 7.1M | 78.3 | 64.5 | 77.2 | 70.8 | 85.0 | 55.5 | 60.7 | 78.4 | 55.4 | 69.5 |
| **Unified** | OMG-LLaVA | 7B | 1.2M | 56.3 | 37.6 | 54.6 | 39.9 | 65.2 | 36.2 | 32.7 | 18.3 | 28.0 | 41.0 |
| | MIMO* | 7B | - | 58.8 | - | 57.0 | - | 52.4 | - | - | - | - | - |
| | MedPLIB | 14B/7B | 500K | 58.3 | 34.8 | 48.4 | 37.4 | 35.2 | 18.2 | 28.7 | 61.9 | 13.1 | 37.3 |
| | MEDSIGHT | 8B | 72K | **79.9** | **61.4** | **70.9** | **60.2** | **66.3** | **42.6** | **51.3** | **68.9** | **58.9** | **62.3** |

clinical workflows. In real-world practice, a clinician must first identify *what* the abnormality is before segmenting its spatial extent. To reflect this diagnosis-then-grounding process, we introduce a new evaluation task, **Grounded Diagnostic Segmentation (DiagSeg)**, designed to jointly assess a model's diagnostic reasoning and its ability to produce pixel-level predictions.

**Task Definition.** Given a medical image and a diagnostic question, the model must first infer the relevant pathological concept and then produce a corresponding segmentation mask aligned with that diagnosis. For instance, when asked "*What is the abnormality observed on this imaging? Please provide the diagnosis and then segment it.*", the model must respond with a diagnostic judgment first and then ground that diagnosis by generating the corresponding mask.

**Dataset Construction.** DiagSeg is constructed from image–mask–description triplets drawn from the *test sets* of multiple public medical datasets that provide diagnostic annotations and segmentation masks. As detailed in Appendix B.1, these datasets cover diverse imaging modalities (CT, MRI, and X-ray) and are widely used in prior medical image segmentation models (Cheng et al., 2023; Huang et al., 2025). For each triplet, we prompt GPT-5 to synthesize clinically plausible diagnostic QA pairs, *strictly constrained* by the provided mask and description. The final evaluation set contains **1,655** VQA pairs, each aligned with a pixel-level segmentation mask. Prompt details, configurations, and human evaluation are reported in Appendix B.

# 4. Experiments

## 4.1. Data and Model Settings

**Training Data.** For the pre-training of the Region Perceiver, we adopt the training set of BiomedParse (Zhao et al., 2025) as $\mathcal{D}_{seg}$. For the Vision-to-Text Alignment stage, we use the PubMedVision (Chen et al., 2024a) alignment dataset containing 647K image–text pairs as $\mathcal{D}_{v \to t}$. For the Text-to-Vision Alignment stage, we construct a medical grounding dataset $\mathcal{D}_{t \to v}$ with 60K samples that include codebook tokens for mask grounding supervision. Finally, we randomly sample 60K instruction–tuning samples from PubMedVision as $\mathcal{D}_{inst}^{r}$ and curate an additional 12K codebook-based instruction–tuning samples as $\mathcal{D}_{inst}^{g}$ to jointly optimize comprehension and segmentation abilities. Detailed training data preparation is provided in Appendix C.

**Implementation Details.** MEDSIGHT builds on Qwen3-8B (Yang et al., 2025) and adopts Unimed-CLIP (ViT-L-14) (Khattak et al., 2024) as the visual encoder. More implementation details are included in Appendix D.

## 4.2. Medical Visual Comprehension

**Evaluation Datasets and Metrics.** As most Med-LVLMs focus on visual comprehension only, we first evaluate MEDSIGHT on diverse medical visual comprehension tasks using standard medical VQA benchmarks including SLAKE (Liu et al., 2021), VQA-RAD (Lau et al., 2018), PathVQA (He et al., 2020), OmniMedVQA (Hu et al., 2024),

*Table 3.* Comparison of MEDSIGHT with other LVLMs and unified multi-modal models on medical diagnosis segmentation task (DiagSeg). **Bold** and underlined text indicate the best performance and second-best performance, respectively. We use abbreviated names for imaging modalities; full names are provided in Appendix E.

| Model | # Params | DiagSeg-Diagnosis ↑ | | | | | | | | | DiagSeg-Segmentation ↑ | | | | | | | | |
|---|---|---|---|---|---|---|---|---|---|---|---|---|---|---|---|---|---|---|---|
| | | CT | MRI | X-ray | Path | US | End | Der | OCT | Avg. | CT | MRI | X-ray | Path | US | End | Der | OCT | Avg. |
| LISA | 7B | 11.8 | 7.9 | 3.8 | 21.6 | 5.9 | 2.9 | 33.4 | 27.8 | 14.1 | 15.7 | 9.8 | 34.0 | 45.2 | 39.3 | 28.9 | 62.6 | 19.1 | 31.8 |
| LISA++ | 7B | 9.3 | 19.6 | 5.1 | 16.7 | 6.2 | 19.4 | 35.7 | 16.7 | 15.1 | 1.9 | 12.5 | 21.6 | 26.9 | 8.9 | 6.8 | 31.2 | 0.7 | 13.8 |
| LaSagnA | 7B | 3.6 | 4.1 | 1.0 | 8.2 | 5.1 | 0.0 | 14.8 | 2.8 | 4.4 | 9.3 | 8.7 | 18.2 | 30.0 | 14.1 | 18.0 | 34.9 | 12.6 | 18.2 |
| GLaMM | 7B | 11.4 | 20.6 | 6.9 | 16.6 | 38.9 | 10.6 | 22.3 | 3.1 | 16.9 | 1.6 | 4.5 | 16.6 | 22.8 | 9.9 | 5.6 | 26.3 | 0.8 | 11.0 |
| OMG-LLaVA | 7B | 12.7 | 23.0 | 10.7 | 17.0 | 34.6 | 5.5 | 26.9 | 15.7 | 18.3 | 0.9 | 5.1 | 15.5 | 20.7 | 11.7 | 7.7 | 25.9 | 1.0 | 11.1 |
| MedPLIB | 14B/7B | 4.5 | 18.5 | 15.3 | 5.3 | 71.9 | 22.4 | 29.0 | 0.4 | 13.1 | 3.8 | 47.1 | 29.8 | 19.3 | 49.9 | 16.6 | 80.4 | 7.2 | 31.8 |
| MEDSIGHT | 8B | **54.6** | **54.4** | **59.1** | **80.1** | 51.7 | **62.4** | **40.2** | **69.3** | **58.9** | **65.8** | **82.3** | **57.7** | **52.9** | **65.8** | **70.0** | **87.7** | **55.6** | **69.9** |

the diagnosis subset of DiagSeg, and the medical subset of MMMU (Yue et al., 2024). Following the LLaVA-Med evaluation protocol (Li et al., 2023a), we report Accuracy for close-ended questions and Recall for open-ended questions. Dataset statistics and details are provided in Appendix E.

**Baselines.** For the visual comprehension task, we compare MEDSIGHT with state-of-the-art general and medical LVLMs. Specifically, we group baselines into two categories: (1) *Comprehension-only LVLMs*: including *general* LVLMs such as BLIP-2 (Li et al., 2023b), LLaVA (Liu et al., 2023), InstructBLIP (Dai et al., 2023), Yi-VL (Young et al., 2024), InternVL2 (Chen et al., 2024b), Llama-3.2 (Dubey et al., 2024), Qwen3-VL and InternVL3.5 (Wang et al., 2025), as well as *medical* LVLMs including Med-Flamingo (Moor et al., 2023), LLaVA-Med (Li et al., 2023a), HuatuoGPT-Vision (Chen et al., 2024a), HealthGPT (Lin et al., 2025), and Lingshu-7B (Xu et al., 2025). (2) *Unified comprehension–segmentation LVLMs*: which include representative *general* model OMG-LLaVA (Zhang et al., 2024), as well as *medical* models including MIMO (Chen et al., 2025) and MedPLIB (Huang et al., 2025). We exclude VividMed (Luo et al., 2024), as it is primarily trained on MIMIC-CXR and is not directly comparable in our evaluation setting.

**Results.** As shown in Table 2, MEDSIGHT achieves superior performance with an average score of 62.3, outperforming the best baseline HuatuoGPT-Vision (58.3) despite being fine-tuned with only 72K instruction-tuning samples, while HuatuoGPT-Vision was fine-tuned with 647K instruction tuning samples. The advantage is even more pronounced among unified multimodal models (e.g., MedPLIB), underscoring the effectiveness of our method and training design. Models highlighted in gray indicate partial overlap between their training and evaluation datasets. For example, HealthGPT and InternVL3.5 include VQA-RAD, SLAKE, and PathVQA for training. Such overlaps make direct comparison less fair but we still report these results for reference and completeness. Even under this setting, MEDSIGHT achieves comparable or even superior performance (e.g.,the VQA-RAD-close), demonstrating its strong comprehension and generalization capability. We further verify that MED-SIGHT retains its advantage under downstream parameter-

*Table 4.* Performance on MeCoVQA-G. While performance of MEDSIGHT on a few OOD datasets is lower, MEDSIGHT still demonstrates cross-modality generalization and achieves the highest average score across modalities.

| Model | Der | CT | PET | X-Ray | End | MR | US | FP | Avg. ↑ |
|---|---|---|---|---|---|---|---|---|---|
| LISA | 45.6 | 7.3 | 4.4 | 4.2 | 30.0 | 9.3 | 16.4 | 6.5 | 15.5 |
| LISA++ | 32.0 | 4.2 | 2.3 | 2.1 | 8.8 | 6.3 | 7.2 | 3.4 | 8.3 |
| LaSagnA | 33.6 | 5.0 | 2.3 | 2.6 | 18.0 | 8.7 | 14.1 | 4.7 | 11.1 |
| GLaMM | 33.5 | 2.6 | 1.0 | 2.2 | 9.6 | 3.2 | 4.1 | 2.9 | 7.8 |
| OMG-LLaVA | 23.6 | 3.5 | 0.8 | 2.2 | 11.7 | 2.7 | 5.5 | 3.5 | 6.7 |
| BiomedParse | 69.6 | 40.5 | 18.8 | 25.4 | 66.2 | 20.8 | 8.8 | 72.0 | 40.3 |
| MedPLIB | **79.8** | **57.2** | **65.6** | 6.6 | 46.1 | **26.3** | **34.4** | 5.1 | 40.1 |
| MEDSIGHT | 71.7 | 46.4 | 22.1 | **26.7** | **70.8** | 23.1 | 9.2 | **72.1** | **42.8** |

efficient fine-tuning on SLAKE, VQA-RAD, and PathVQA against medical LVLMs of comparable scale; full results are reported in Appendix F.4.

### 4.3. Grounded Diagnostic Segmentation

**Evaluation Datasets and Metrics.** After evaluation of basic visual comprehension ability, we further evaluate MED-SIGHT on a joint visual comprehension and segmentation task using the proposed **DiagSeg** benchmark, which measures a model's ability to perform diagnostic reasoning with pixel-level grounding. We report the mean Dice score for segmentation performance and, following LLaVA-Med (Li et al., 2023a), use Recall to evaluate diagnostic answers.

**Baselines.** As *Comprehension-only LVLMs* do not support this joint task, we compare MEDSIGHT with representative *unified comprehension–segmentation LVLMs*, including OMG-LLaVA and MedPLIB. We further include segmentation-oriented unified models LISA (Lai et al., 2024), LISA++ (Yang et al., 2023), LaSagnA (Wei et al., 2024) and GLaMM (Rasheed et al., 2024). We exclude MIMO, as its model and data are not publicly available.

**Results.** As shown in Table 3, MEDSIGHT achieves the best overall performance in both diagnostic reasoning and pixel-level grounding across eight medical imaging modalities. On *DiagSeg-Diagnosis*, it attains an average Recall of 58.9, substantially outperforming all baselines. While MEDSIGHT underperforms MedPLIB on the US modality, we provide a detailed analysis in Appendix F.1. In contrast, segmentation-oriented models (e.g., LISA) struggle to jointly support comprehension and segmentation, often producing segmentation tokens only. On *DiagSeg-*

*Table 5.* Performance of ablation study.

| With $Q_r$ | With $\mathcal{C}$ | Align $v \to t$ | Align $t \to v$ | Unified Tuning | VQA-RAD close | all | SLAKE close | all | PathVQA close | all | MMMU-Med | OMVQA | DiagSeg-VQA | DiagSeg-Seg | MeCo |
|---|---|---|---|---|---|---|---|---|---|---|---|---|---|---|---|
| ✓ | ✓ | ✓ | ✓ | ✓ | 79.9 | 61.4 | 70.9 | 60.2 | 66.3 | 42.6 | 51.3 | 68.9 | 58.9 | 69.9 | 42.8 |
| ✗ | ✓ | ✓ | ✓ | ✓ | 72.8 | 59.4 | 66.8 | 57.9 | 62.6 | 40.7 | **54.7** | 65.3 | 54.4 | 59.2 | 37.6 |
| ✓ | ✗ | ✓ | ✗ | ✓ | 77.2 | 61.0 | 66.3 | 58.5 | 63.7 | 40.8 | 50.0 | 67.0 | 56.4 | 60.4 | 40.0 |
| ✓ | ✓ | ✗ | ✓ | ✓ | 73.6 | 56.4 | 57.9 | 53.1 | 64.4 | 40.9 | 49.3 | 65.5 | 56.6 | 63.2 | 42.7 |
| ✓ | ✓ | ✓ | ✗ | ✓ | 74.0 | 60.3 | 66.3 | 57.9 | 65.4 | 42.3 | 50.0 | 66.9 | 58.7 | 59.4 | 40.6 |
| ✓ | ✓ | ✓ | ✓ | ✗ | 44.9 | 41.6 | 43.5 | 44.2 | 29.3 | 24.5 | 31.3 | 44.6 | 45.1 | - | - |

*Segmentation*, MEDSIGHT also leads with a mean Dice of 69.9, surpassing both medical and general-domain segmentation LVLMs. Despite relying on heavily pre-trained backbones (e.g., SAM-Med2D), existing methods exhibit weaker performance, highlighting the effectiveness of our unified reasoning–grounding framework.The framework also generalizes beyond segmentation to bounding-box grounding, achieving the best zero-shot mIoU on S-Chain and MedTrinity-25M (Appendix F.5).

### 4.4. Text-Prompted Medical Image Segmentation

**Evaluation Datasets and Metrics.** MEDSIGHT also supports simple text-prompted segmentation. We evaluate this capability on the MeCoVQA-G benchmark, which covers diverse imaging modalities and includes several out-of-distribution (OOD) cases for MEDSIGHT. Performance is measured using the mean Dice score.

**Baselines.** We adopt *unified comprehension–segmentation LVLMs* in Sec. 4.3 as baselines and additionally include the state-of-the-art medical segmentation model Biomed-Parse (Zhao et al., 2025). We exclude the MedSAM series (Ma et al., 2024; 2025) due to their reliance on visual prompts, and MedSAM3 (Liu et al., 2025) as it is not yet publicly available.

**Results.** As shown in Table 4, MEDSIGHT achieves the highest overall performance with 42.8, surpassing MedPLIB even though it is partially trained on the MeCoVQA-G training set. While performance on a few OOD datasets is lower, MEDSIGHT still demonstrates strong cross-modality generalization and achieves the highest average score.

### 4.5. Ablation Study

We perform a comprehensive ablation study to assess the contribution of each core component in MEDSIGHT, as summarized in Table 5. Our analysis focuses on three major aspects: region embeddings, codebook integration, and training design.

**Effect of Region Embeddings.** To test the effect of Region Perceiver $\mathcal{R}$, we remove the region embeddings $\mathbf{Q}_r$ from the LLM input, which leads to a significant drop in performance across almost all datasets, highlighting the necessity of incorporating fine-grained information in Med-LVLMs.

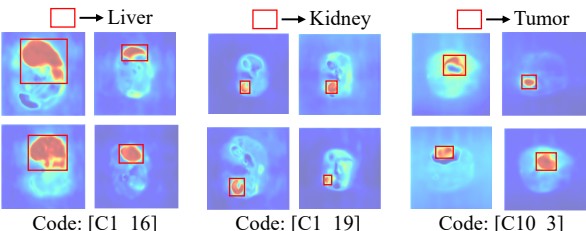

*Figure 3.* Visualization of representative codes from our trained region codebook. Each code is illustrated using four segmentation logit maps that reflect its learned spatial semantics.

A supervision-matched variant that retains the same segmentation pretraining but replaces $\mathcal{R}$ with a simpler one-way query decoder also underperforms MEDSIGHT, isolating the architectural contribution of the Region Perceiver (Appendix F.7).

**Effect of Codebook Integration.** To evaluate the role of the region codebook $\mathcal{C}$, we replace it with only one single `[SEG]` token. This simplification causes substantial degradation, particularly on segmentation tasks, highlighting the importance of diverse and fine-grained region tokens. To understand **why the region codebook is effective**, we visualize the reconstructed segmentation logits of several representative codes. As shown in Figure 3, each code captures a distinct grounding pattern with clear anatomical semantics. For example, logits of code `[C1_16]` primarily attends to the liver region (highlighted with red boxes) in abdominal CT. These results demonstrate that the learned codes encode meaningful grounding priors, explaining their effectiveness within our unified reasoning and segmentation framework. Additional results of codebook-size ablation are reported in Appendix F.8.

**Effect of Alignment and Unified Tuning.** We further ablate each stage in the training pipeline. (1) Removing the vision-to–text ($v \to t$) alignment in Sec. 2.2 disrupts the projection of visual embeddings $\mathcal{P}_{v \to t}$, leading to poor cross-modal alignment and degraded performance on all benchmarks. (2) Omitting the text-to-vision alignment ($t \to v$) in Sec. 2.3, i.e., the codebook grounding stage, prevents effective mapping between region codes and their visual counterparts, resulting in a large drop in comprehension and segmentation metrics. (3) Excluding the unified grounding instruction tuning stage (Sec. 2.4) severely impairs both instruction following and grounding ability. Without this stage, the model

even fails to output codebook tokens for segmentation.

To rule out backbone capacity as a confounder, we further train a 7B variant of MEDSIGHT (Qwen2.5-7B) that still outperforms comparable medical LVLMs on both VQA-RAD and DiagSeg (Appendix F.6).

## 5. Related Work

**Large Vision-Language Models.** Recent medical LVLMs (Moor et al., 2023; Li et al., 2023a; Chen et al., 2024a; Lin et al., 2025; Xu et al., 2025; Chang et al., 2025; 2026) achieve strong semantic understanding but remain limited in pixel-level grounding for precise localization. LISA (Lai et al., 2024) and subsequent methods (Yang et al., 2023; Wei et al., 2024) pioneered reasoning-based segmentation by integrating SAM with LLMs, where special tokens activate segmentation modules. This paradigm has since been extended by unified models such as GLaMM (Rasheed et al., 2024) and OMG-LLaVA (Zhang et al., 2024) to support multi-granular visual grounding and comprehension. Medical unified models (Huang et al., 2025; Luo et al., 2024; Chen et al., 2025) further adapt this approach for grounded medical visual understanding. However, these architectures still lack fine-grained spatial inputs for grounding and rely on a single special token to represent all segmentation targets, which limits their expressive capacity.

**Medical Visual Grounding Datasets.** Prior to DiagSeg, several medical visual grounding datasets have been proposed. BiomedParse (Zhao et al., 2025) integrates multiple public medical segmentation datasets, which largely overlap with the collections in SA-Med2D-20M (Ye et al., 2023). MedPLIB (Huang et al., 2025) further proposed MeCoVQA-Grounding for simple text-prompted medical segmentation. More recently, MIMO (Chen et al., 2025) introduced MIMO-Seg, a non-public dataset focusing on organ segmentation and grounding with both simple and complex text prompts. However, most existing datasets directly provide segmentation targets, whereas in real clinical workflows, such targets often require diagnostic inference and may not be explicitly available. DiagSeg is therefore proposed to address this gap by enabling comprehensive evaluation of a model's joint diagnostic comprehension and pixel-level grounding ability.

**Visual Grounding Models.** Our Region Perceiver is inspired by early work DETR (Carion et al., 2020) and Mask2Former (Cheng et al., 2022). Mask2Former employs separate pixel and mask decoders, where region tokens attend to visual features via masked cross-attention and self-attention, without feedback from visual features. In contrast, our Region Perceiver unifies pixel- and region-level processing within a single module using dual cross-attention, enabling bidirectional interaction and learnable feature up-scaling. In the medical domain, SAM (Kirillov

et al., 2023) and its variants MedSAM (Ma et al., 2024) and MedSAM2 (Ma et al., 2025) focus on visually prompted segmentation, while the recent MedSAM3 (Liu et al., 2025) additionally supports text prompts. BiomedParse (Zhao et al., 2025), built upon the SEEM architecture (Zou et al., 2023), achieves state-of-the-art performance in text-prompted medical segmentation. However, these methods primarily emphasize segmentation and largely overlook text reasoning and comprehension. We address this gap by proposing a unified model, MEDSIGHT, that jointly supports visual grounding and semantic reasoning.

## 6. Conclusions

We introduce MEDSIGHT, a unified architecture that endows Med-LVLMs with structured, pixel-level understanding for both visual comprehension and segmentation. By pretraining a Region Perceiver and a modality-aware Region Codebook, and integrating them through grounded instruction tuning, MEDSIGHT achieves strong and interpretable performance diverse imaging modalities. Future work will explore improving codebook generalization by introducing more flexible designs, such as template or void codes, to better handle unseen modalities and regions.

## Acknowledgements

We thank the anonymous reviewers for their insightful comments. We also thank Dr. Anjali Rajeev from GE Health-Care as one of the experts for participating in the human validation experiments and Dr. Haidong Yi from St. Jude Children's Research Hospital for providing valuable suggestions and feedback on paper writing in the Introduction and Methodology sections. This work was also partially supported by the National Science Foundation under Grant No. 223827 (F. Ma).

## Impact Statement

This paper presents a method for grounded visual comprehension in medical vision–language models, aiming to improve reliability and interpretability. By reducing misgrounding, our work may support safer multimodal medical AI systems. As with all machine learning approaches in healthcare, the proposed method is not intended to replace professional medical judgment. Errors, hallucinations, biases, or failures to generalize to unseen modalities may still occur. Our models are trained on existing de-identified datasets, and any biases present in these data may propagate to model behavior. Overall, this work advances medical multimodal learning while emphasizing the importance of grounding, careful evaluation, and responsible use in high-stakes settings.

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

# A. Detailed Training Objectives

## A.1. Region Perceiver

Given the final set of region queries $\mathbf{Q}_r = \{\mathbf{q}_i\}_{i=1}^N, \mathbf{q}_i \in \mathbb{R}^d$, and the highest–resolution visual feature map after upsampling $\mathbf{I}_r \in \mathbb{R}^{H^L \times W^L \times d}$, the Region Perceiver predicts a segmentation mask for each region by computing the dot product between $\mathbf{q}_i$ and the spatial feature vectors at all positions:

$$\hat{\mathbf{M}}_i(\mathbf{I}_r, \mathbf{q}_i) = \sigma(\mathbf{I}_r \mathbf{q}_i), \qquad \hat{\mathbf{M}}_i \in [0, 1]^{H^L \times W^L}, \tag{2}$$

where $\sigma(\cdot)$ denotes the sigmoid activation.

**Ground-truth masks and interpolation.** For each region, the dataset provides a ground-truth binary mask $\mathbf{M}_i^{\text{gt}} \in \mathbb{R}^{H^* \times W^*}$. Collectively, the full supervision mask tensor is $\mathbf{M}^{\text{gt}} = \{\mathbf{M}_i^{\text{gt}}\}_{i=1}^N \in \mathbb{R}^{N \times H^* \times W^*}$. Since predicted masks are produced at resolution $(H^L, W^L)$, they are first interpolated to the target resolution via bilinear upsampling:

$$\tilde{\mathbf{M}}_i(\mathbf{I}_r, \mathbf{q}_i) = \texttt{Interp}\Big(\hat{\mathbf{M}}_i(\mathbf{I}_r, \mathbf{q}_i),\, H^*,\, W^*\Big), \tag{3}$$

yielding the final mask prediction set:

$$\tilde{\mathbf{M}}(\mathbf{I}_r, \mathbf{Q}_r) = \{\tilde{\mathbf{M}}_i(\mathbf{I}_r, \mathbf{q}_i)\}_{i=1}^N \in \mathbb{R}^{N \times H^* \times W^*}.$$

**Segmentation losses.** Following DETR (Carion et al., 2020), MaskFormer (Cheng et al., 2021) and Mask2Former (Cheng et al., 2022), we use the Binary Cross-Entropy loss:

$$\mathcal{L}_{\text{bce}}(\mathbf{I}_r, \mathbf{Q}_r) = \text{BCE}\Big(\tilde{\mathbf{M}}(\mathbf{I}_r, \mathbf{Q}_r),\, \mathbf{M}^{\text{gt}}\Big). \tag{4}$$

Similarly, the Dice loss is also included:

$$\mathcal{L}_{\text{dice}}(\mathbf{I}_r, \mathbf{Q}_r) = \text{Dice}\Big(\tilde{\mathbf{M}}(\mathbf{I}_r, \mathbf{Q}_r),\, \mathbf{M}^{\text{gt}}\Big). \tag{5}$$

The combined segmentation objective is:

$$\mathcal{L}_{\text{seg}}(\mathbf{I}_r, \mathbf{Q}_r) = \lambda_1 \, \mathcal{L}_{\text{bce}}(\mathbf{I}_r, \mathbf{Q}_r) + \lambda_2 \, \mathcal{L}_{\text{dice}}(\mathbf{I}_r, \mathbf{Q}_r). \tag{6}$$

**Region-level classification loss.** To endow region queries with semantic meaning, given the ground truth class label $\mathbf{y}_i^{\text{gt}}$, we employ a lightweight classification head $\mathcal{D}_{\text{cls}}$ to predict a class label for each region query token:

$$\hat{\mathbf{y}}_i = \mathcal{D}_{\text{cls}}(\mathbf{q}_i), \qquad \mathcal{L}_{\text{ce}}(\mathbf{Q}_r) = \frac{1}{N} \sum_{i=1}^N \text{CE}\big(\hat{\mathbf{y}}_i, \mathbf{y}_i^{\text{gt}}\big). \tag{7}$$

Following DETR (Carion et al., 2020), Hungarian matching is used to establish a one-to-one assignment between predicted and ground-truth regions during training.

**Overall objective.** The full Region Perceiver pre-training objective is:

$$\mathcal{L}_{\mathcal{R}}(\mathbf{I}_r, \mathbf{Q}_r) = \mathcal{L}_{\text{seg}}(\mathbf{I}_r, \mathbf{Q}_r) + \mathcal{L}_{\text{ce}}(\mathbf{Q}_r). \tag{8}$$

This learning objective equips the Region Perceiver with spatially grounded, semantically rich region representations, forming a strong foundation for downstream multimodal reasoning in MEDSIGHT.

## A.2. Codebook

The Region Perceiver produces a set of region embeddings $\mathbf{Q}_r = \{\mathbf{q}_i\}_{i=1}^N$, where $\mathbf{q}_i \in \mathbb{R}^d$. To obtain a discrete and modality-aware representation space, we apply vector quantization to map each region embedding to the closest codebook item.

**Vector quantization loss.** Following VQ-VAE (Van Den Oord et al., 2017; Razavi et al., 2019), the vector quantization loss contains two components: (i) a *codebook loss* that updates the code vectors toward the encoder outputs, and (ii) a *commitment loss* that encourages the encoder to commit to the selected code and prevents codebook collapse. Using the stop-gradient operator $\text{sg}(\cdot)$, the VQ loss is:

$$\mathcal{L}_{\text{VQ}} = \left\| \text{sg}[\mathbf{Q}_r] - \tilde{\mathbf{Q}}_r \right\|_2^2 + \beta \left\| \mathbf{Q}_r - \text{sg}[\tilde{\mathbf{Q}}_r] \right\|_2^2, \tag{9}$$

where $\tilde{\mathbf{Q}}_r = \{\tilde{\mathbf{q}}_i\}_{i=1}^N$ is the set of quantized region tokens, and $\beta$ is the commitment weight. Following VQ-VAE, we set $\beta$ as 0.25.

**Reconstruction loss.** To preserve the fine-grained semantics encoded in the original region embeddings, we add an $\ell_2$ reconstruction loss:

$$\mathcal{L}_{\text{recon}} = \left\| \tilde{\mathbf{Q}}_r - \mathbf{Q}_r \right\|_2^2. \tag{10}$$

**Spatial grounding preservation.** To maintain consistency with the spatial grounding learned by the Region Perceiver, the original Region Perceiver objective $\mathcal{L}_\mathcal{R}$ (Eq. (1)) is reused by substituting region queries with their quantized counterparts $\tilde{\mathbf{Q}}_r$:

$$\mathcal{L}_\mathcal{R}\big(\mathbf{I}_r, \tilde{\mathbf{Q}}_r\big) = \mathcal{L}_{\text{seg}}\big(\mathbf{I}_r, \tilde{\mathbf{Q}}_r\big) + \mathcal{L}_{\text{ce}}\big(\tilde{\mathbf{Q}}_r\big). \tag{11}$$

**Final training objective.** The complete codebook training loss combines vector quantization, reconstruction, and grounding preservation:

$$\mathcal{L}_{\text{codebook}} = \mathcal{L}_{\text{VQ}} + \mathcal{L}_{\text{recon}} + \mathcal{L}_\mathcal{R}(\mathbf{I}_r, \tilde{\mathbf{Q}}_r). \tag{12}$$

This training objective enables the codebook to capture modality-consistent visual patterns while preserving the spatial grounding ability of the Region Perceiver. Notably, although the codebook loss contains three components, we do not introduce additional weighting coefficients. In practice, we find that using equal weighting yields stable codebook learning and accurate segmentation reconstruction, which is sufficient for downstream LLM integration. Exploring more sophisticated weighting strategies is left for future work.

---

**Prompt for DiagSeg dataset construction**

**System Message:**
You are an expert radiologist and dataset curator. You are given region-level annotations for one or more medical images. Each image item includes: "image_id": an integer id, "masks": a list of region objects, each with: "id": id of this mask "sentences": free-text descriptions (list of strings). Assume you can see the image implicitly and must use only the provided annotation information (sentences) to perform the tasks below.

**Task** (produce a single JSON output per image):
• Generate evaluation data in the form of VQA about the image and segmentation based on the provided region information.
• The generated data should target at evaluating diagnosis and segmentation concurrently.
• Follow the style of radiology and clinical reasoning, and make sure the generated questions are natural, factual, and unambiguous.

**QA generation rules:**
• For each image, generate one diagnostic Q&A pair for each provided mask.
• Each QA serves with 2 evaluation targets: VQA and segmentation.
• For each QA, include a short version of answer for the convenience of evaluation VQA (short diagnosis for open-ended VQA).
• For the mask with only organ and without abnormalities, you should skip it by generating empty QAs.
• For the mask with abnormalities, you may use open-ended questions to ask the diagnosis.

*Figure 4.* Prompt used for constructing the DiagSeg dataset.

## B. Grounded Diagnostic Segmentation Dataset

### B.1. Source Datasets

To build DiagSeg for grounded diagnosis and segmentationWe use a subset of the BiomedParse dataset (Zhao et al., 2025), which provides abnormal regions annotated with textual descriptions and segmentation masks. Specifically, we sample image–mask–description triples from the following datasets: Breast Ultrasound (Al-Dhabyani et al., 2020), ISIC (Codella et al., 2019), CESM (Khaled et al., 2022), KiTS23 (Heller et al., 2023), MSD (Antonelli et al., 2022), LIDC-IDRI (Armato III et al., 2011), NeoPolyp (Ngoc Lan et al., 2021), OCT-CME (Ahmed et al., 2022), PanNuke (Gamper et al., 2020),

---

**Prompt for unified instruction tuning data generation**

**System Message:**
You are an expert radiologist and dataset curator. You are provided with region-level annotations for one or more medical images. Each image item includes the following fields: "image_id": integer identifier of the image; "masks": a list of region objects, each containing: (1) "quantizer_code": unique identifier for this region (e.g., "M1_25"). (2) "sentences": free-text region descriptions (list of strings). Assume you can see the image implicitly and must use only the provided annotation information (sentences) to perform the tasks below.

**Task**
For each input image, generate user–assistant dialogues that demonstrate how a model should reason about the image, focusing on segmentation, detection, and localization tasks.

**Segmentation and Detection Rules**
When the user requests segmentation/detection/localization, the Assistant MUST: (1) Mention the corresponding quantizer codes inline, e.g., "tumor [M1_25]". (2) Include "mask_ids_order": a list of 0-based indices into the "masks" list, in the same order the Assistant refers to them.

**Dialogue Composition Guidelines**
(1) Generate 2-4 dialogue rounds per image. (2) The first dialogue should naturally introduce the overall task context, using a realistic and conversational style (not a meta or dataset tone). (3) After the introduction, you should only include those segmentation-related user intents, such as: Direct segmentation/detection requests ("Please segment the tumor."). (4) If there are multiple masks, prioritize clinically significant regions (e.g., tumors, lesions, major organs).

*Figure 5.* Prompt used for constructing the instruction–tuning dataset $\mathcal{D}_{\text{inst}}^g$.

COVID-19-CT (Jun et al., 2020), QaTa-COV19 (Degerli et al., 2022), SIIM-ACR Pneumothorax Segmentation[3], and UwaterlooSkinCancer [4]. The source datasets cover diverse medical imaging modalities as shown in the main experiment results. Note that existing unified medical LVLMs, including MEDSIGHT and MedPLIB, partially overlap with these datasets during training; for example, MedPLIB is trained on samples from SA-Med2D-20M, which include most of these datasets.

## B.2. Dataset Construction

Using the image–mask–description triples, GPT-5[5] generates diagnostic questions based on the diagnostic information present in the region descriptions. We include the prompt used for data generation in Figure 4.

## B.3. Human Validation

Before using the GPT-5 generated test data for quantitative evaluation, we conducted a human validation study to ensure the reliability of the generated questions and answers. We recruited a licensed physician to carefully examine a randomly selected subset of 100 image–question–answer triplets sampled from all 1,655 testing cases. The sampled examples span a diverse set of radiological imaging techniques, including ultrasound, CT, X-ray, and others.

During validation, the physician cross-referenced the images, corresponding segmentation masks, and the generated textual descriptions, and then assigned a consistency score on a 1–5 scale. The results demonstrate excellent quality: 98% of the generated answers received a perfect consistency score of 5, while the remaining 2% received a score of 3, yielding an average score of 4.96/5.00 across all validated examples. This high level of agreement indicates that GPT-5 produces testing data with strong clinical and visual consistency, meeting our expectations for downstream evaluation. Based on this validation, we proceed to use the generated test set for all subsequent experiments.

**Extended multi-rater validation with modality-balanced sampling.** To further strengthen the reliability of DiagSeg, we conducted an extended validation study with an *independent* second medical expert (Expert B) in addition to the original annotator (Expert A). To ensure balanced coverage across all eight imaging modalities (with very different diagnostic characteristics), we sampled 30 cases per modality, resulting in a modality-balanced validation set of 240 cases in total. Each expert independently assigned a 1–5 consistency score to every case, and we report inter-rater agreement using both Pearson and Spearman correlation coefficients. The modality-level results are summarized in Table 6.

Across all 240 samples, Expert A and Expert B achieved mean consistency scores of 4.77 and 4.79, respectively, indicating consistently high agreement with the GPT-5–generated diagnostic content. The high Pearson and Spearman correlations on the modalities with non-degenerate score variance (Dermatoscopy, MRI, Pathology, X-Ray) further demonstrate strong agreement between independent annotators across diverse diagnostic characteristics. For the remaining four modalities (CT,

---

[3] https://www.kaggle.com/datasets/vbookshelf/pneumothorax-chest-xray-images-and-masks
[4] https://vip.uwaterloo.ca/skin-cancer-detection/
[5] OpenAI GPT-5 model, accessed via https://openai.com.

*Table 6.* Extended human validation of DiagSeg with two independent medical experts across all eight imaging modalities. We report mean consistency scores per expert as well as Pearson and Spearman correlations. "n/a" indicates that the correlation is undefined because one or both raters have zero variance in that modality (all scores are identical).

| Modality | Expert A Mean | Expert B Mean | Pearson $r$ | Spearman $r$ |
|---|---|---|---|---|
| CT | 4.90 | 5.00 | n/a | n/a |
| Dermatoscopy | 4.13 | 4.20 | 0.869 | 0.896 |
| Endoscopy | 5.00 | 5.00 | n/a | n/a |
| MRI | 4.80 | 4.67 | 0.775 | 0.777 |
| OCT | 4.97 | 5.00 | n/a | n/a |
| Pathology | 4.83 | 4.90 | 0.745 | 0.745 |
| Ultrasound | 5.00 | 5.00 | n/a | n/a |
| X-Ray | 4.50 | 4.53 | 0.935 | 0.935 |
| **Overall (240 cases)** | **4.77** | **4.79** | – | – |

Endoscopy, OCT, Ultrasound), one or both raters consistently assigned the same score across nearly all cases, leaving the correlation undefined; in these cases the very high (often perfect) per-modality means already directly indicate excellent annotation quality. Together, these multi-rater results support the reliability of DiagSeg as a benchmark for grounded diagnostic evaluation.

## C. Training Data Settings

During the final Unified Grounded Instruction Tuning stage, we randomly sample 60K instruction–tuning examples from PubMedVision to form $\mathcal{D}_{\text{inst}}^r$ and curate an additional 12K codebook-based instruction–tuning samples as $\mathcal{D}_{\text{inst}}^g$. These two datasets are used jointly to optimize both visual comprehension and segmentation capabilities. Below, we describe the data preparation procedure used to construct these instruction–tuning samples based on our pretrained Region Perceiver and GPT-5-mini.

To generate $\mathcal{D}_{\text{inst}}^g$, we uniformly sample 12K images from the pool of BiomedParse with more than 250K images spanning diverse modalities. For each sampled image, we first extract region codes using the pretrained Region Perceiver and the learned region codebook, and append these codes to the corresponding image annotations. Each annotation therefore contains the image, its segmentation masks, region-level text descriptions, and the associated discrete codes. These enriched annotations are then provided to the LLM to generate multi-turn conversations for grounded instruction tuning. The complete prompt used for GPT-5-mini is provided in Figure 5.

## D. Implementation Details

MEDSIGHT is built on the pretrained LLM Qwen3-8B (Yang et al., 2025) and uses UniMed-CLIP (ViT-L/14) (Khattak et al., 2024) as the visual encoder. For the Region Perceiver, we set the number of layers to $L = 3$ and use 20 region query tokens. Pre-training is conducted on the BiomedParse training set for 20 epochs with a learning rate of $1 \times 10^{-4}$. In the segmentation loss $\mathcal{L}_\mathcal{R}$, we specify the hyperparameters $\lambda_1$ and $\lambda_2$ both as 5 following Mask2Former (Cheng et al., 2022). The codebook is trained using the pretrained region embeddings from the Region Perceiver, also on BiomedParse, for 3 epochs with a learning rate of $1 \times 10^{-4}$. The number of modalities is set to $K = 18$ following the fine-grained modality categories in BiomedParse. Each codebook vector has dimension $d_c = 64$, and each modality contains $M = 32$ codes. Importantly, the attention from region queries to images, $\text{CrossAtt}_{r \to i}$ is implemented using the masked attention following Mask2Former.

For Vision-to-Text Alignment, only the vision-to-text projector $\mathcal{P}_{v \to t}$ is trainable; we train this stage for 1 epoch with a learning rate of $5 \times 10^{-4}$. For Text-to-Vision Alignment, we train the text-to-vision projector $\mathcal{P}_{t \to v}$ and the LLM-side codebook embeddings for 3 epochs, also with a learning rate of $5 \times 10^{-4}$. Finally, during the instruction tuning stage, we train the LLM and alignment layers for 3 epochs with a learning rate of $2 \times 10^{-5}$. All experiments are conducted on 4×H100 GPUs.

For the GPT-5 and GPT-5-mini models used during DiagSeg dataset and training dataset construction, we employed the versions `beta-gpt-5-2025-08-07` and `beta-gpt-5-mini-2025-08-07`, respectively.

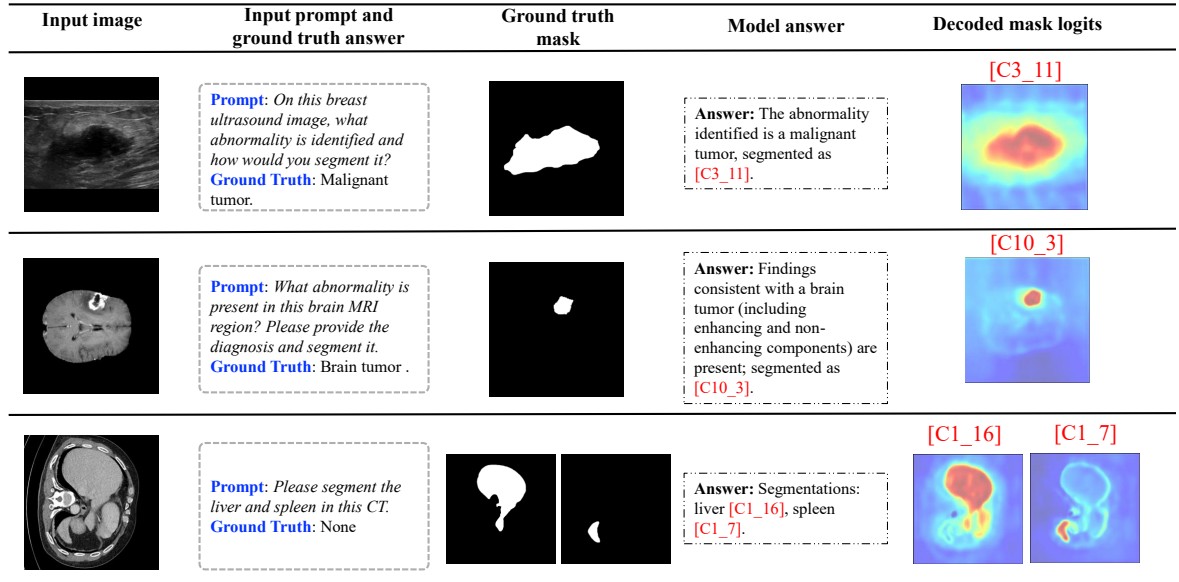

*Figure 6.* Case study of MEDSIGHT, showing generated text responses and the decoded segmentation masks obtained from the predicted region codes.

# E. Evaluation Benchmarks

## E.1. DiagSeg for Joint Evaluation

We use our constructed DiagSeg to do joint evaluation of both medical comprehension and segmentation. DiagSeg covers eight imaging modalities, shown in abbreviated form in the main tables. Their full names are provided here for completeness: Computed Tomography (CT), Magnetic Resonance Imaging (MRI), X-ray Radiography (X-ray), Digital Pathology (Path), Ultrasound (US), Endoscopy (End), Dermatoscopy (Der), and Optical Coherence Tomography (OCT).

## E.2. Visual Comprehension

As introduced in the experimental settings, we evaluate all methods on

*Table 7.* Dataset statistics for the evaluation benchmarks, including the joint comprehension–segmentation (Joint) task, the medical visual comprehension (Comp.) task, and the text-prompted segmentation (Seg.) task.

| Task | Dataset | Data Scale |
|------|---------|-----------|
| Joint | DiagSeg | 1,655 |
| | SLAKE | 1,061 |
| | VQA-RAD | 451 |
| Comp. | PathVQA | 6,761 |
| | MMMU-Med | 180 |
| | OmniMedVQA | 82,316 |
| Seg. | MeCoVQA-G | 4,126 |

a diverse suite of visual comprehension datasets, with dataset statistics summarized in Table 7. Following HealthGPT (Lin et al., 2025), we use the validation split of MMMU-Med for all comparisons. For the OmniMed benchmark, we also adopt the modality selection used in HealthGPT. Specifically, the evaluation covers the following imaging modalities: Computed Tomography, X-ray Radiography, Fundus Photography, Microscopy Imaging, Optical Coherence Tomography, Magnetic Resonance Imaging, and Ultrasound.

## E.3. Text-prompted Segmentation

We evaluate text-prompted segmentation using the MeCoVQA-Grounding (MeCoVQA-G) dataset from MedPLIB (Huang et al., 2025), which contains 4,142 segmentation-focused VQA samples. As part of preprocessing, we remove entries with missing or incorrectly provided masks, following the issues reported in the authors' GitHub repository.[6] The resulting dataset spans eight imaging modalities, which are shown in abbreviated form in the main tables. For clarity, we provide their full names here in the appendix: Dermatoscopy (Der), Computed Tomography (CT), Positron Emission Tomography (PET), X-ray Radiography (X-Ray), Endoscopy (End), Magnetic Resonance Imaging (MR), Ultrasound (US), and Fundus Photography (FP).

---

[6]https://github.com/ShawnHuang497/MedPLIB

# F. Further Analysis

## F.1. Ultrasound Modality Performance Analysis of *DiagSeg-Diagnosis*

While MEDSIGHT achieves the best performance across all other imaging modalitiesin *DiagSeg-Diagnosis*, its comparatively lower score on ultrasound compared with MedPLIB, can be attributed to characteristics of the evaluation data and baseline training data overlap rather than limitations of the proposed method. Specifically, the ultrasound subset in DiagSeg is dominated by breast tumor cases, which strongly overlap with MedPLIB's training set of ultrasound data collected from SA-Med2D-20M. This substantial overlap provides MedPLIB with a favorable prior on diagnostic patterns for this modality.

In contrast, MEDSIGHT is trained under a unified multi-modality setting with less specialization for ultrasound and breast tumor cases. Despite this disadvantage, MEDSIGHT remains competitive on ultrasound and continues to outperform all baselines on the remaining modalities. These results suggest that the observed performance gap is largely driven by dataset bias and training overlap, rather than deficiencies in joint reasoning and grounding. We leave modality-specific adaptation and debiasing strategies as promising directions for future work.

## F.2. Codebook Analysis

In addition to the quanlatitive analysis, we provide a quantitative code distribution analysis in Table 8, complementing the qualitative results in Fig. 3. Using 100 abdominal CT images from the BiomedParse test set, we analyze the Top-3 code assignments for liver and kidney regions. The highly concentrated distributions show that region codes remain semantically consistent across samples rather than being arbitrarily assigned.

*Table 8*. Top-3 region code distribution.

| Anatomy | Rank | Code ID | Frequency |
|---|---|---|---|
| | Top-1 | C1_16 | 98% |
| Liver | Top-2 | C1_7 | 2% |
| | Top-3 | - | 0% |
| | Top-1 | C1_19 | 96% |
| Kidney | Top-2 | C1_11 | 3% |
| | Top-3 | C1_7 | 1% |

## F.3. Case Study

To further demonstrate the capabilities of MEDSIGHT, we present qualitative case studies showing how the model generates region-level codes, segmentation predictions, and diagnostic answers. As illustrated in Figure 6, MEDSIGHT can perform diagnosis while producing region codes that correspond to specific anatomical or pathological regions. These codes can be decoded into segmentation mask logits, which are visualized in the last column.

Beyond handling single-region cases, MEDSIGHT is also capable of generating multiple region codes within one response, enabling multi-region segmentation and diagnosis. As shown in the third example, the model successfully identifies and segments multiple regions simultaneously, without being constrained to a single special token, and delivers precise multi-region segmentation results.

Overall, these case studies highlight the interpretability, flexibility, and consistency of MEDSIGHT across diverse clinical imaging modalities.

## F.4. Downstream Adaptation via LoRA Fine-Tuning

The main paper focuses on zero-shot evaluation, which primarily measures the quality of the learned representations. In realistic clinical scenarios, however, downstream adaptation via fine-tuning is also common due to the domain gap between general pretraining data and specific clinical datasets. To complement the zero-shot results, we conduct an additional study under parameter-efficient fine-tuning (LoRA) settings on three widely used medical VQA benchmarks: SLAKE (Liu et al., 2021), VQA-RAD (Lau et al., 2018), and PathVQA (He et al., 2020). We follow the same protocol as ExGra-Med (MH Nguyen et al., 2026), and restrict comparisons to recent medical multimodal LLMs of comparable scale ($\sim$7–8B parameters) to control for backbone capacity. Specifically, we apply LoRA to all linear layers with rank 32, and fine-tune all models under the same training setting and data splits. We report Accuracy for close-ended VQA and Recall for open-ended VQA. The results are reported in Table 9.

MEDSIGHT consistently outperforms all baselines across the three benchmarks under the same downstream fine-tuning protocol. This indicates that the proposed Region Perceiver and modality-aware codebook do not merely provide strong zero-shot representations: the architecture also retains superior performance when adapted to clinical datasets, supporting its practical relevance for real-world deployment.

*Table 9.* LoRA fine-tuning results on medical VQA benchmarks. All models are fine-tuned under identical settings and data splits using LoRA with rank 32 applied to all linear layers. We report Accuracy for close-ended questions and Recall for open-ended questions.

| Model | SLAKE (close) | SLAKE (open) | VQA-RAD (close) | VQA-RAD (open) | PathVQA (close) | PathVQA (open) |
|---|---|---|---|---|---|---|
| LLaVA-Med 1.5 | 88.5 | 83.3 | 74.4 | 36.7 | 93.2 | 38.0 |
| HuatuoGPT-Vision 7B | 90.1 | 85.6 | 76.8 | 41.2 | 93.3 | 37.1 |
| ExGra-Med | 88.9 | 85.1 | 75.2 | 38.9 | 93.3 | 37.9 |
| MEDSIGHT (ours) | **93.2** | **89.9** | **82.3** | **43.9** | **94.2** | **38.1** |

## F.5. Bounding-Box Grounding Generalization

While the main experiments focus on segmentation-based grounding, bounding-box localization is also a practical and challenging spatial-grounding modality in clinical workflows, especially when lesions visually resemble surrounding tissues. Like unified grounding models such as OMG-LLaVA (Zhang et al., 2024) and MedPLIB (Huang et al., 2025), MEDSIGHT naturally supports bounding-box localization by *deriving bounding boxes from the decoded segmentation masks*, which is a standard practice in grounding-based frameworks.

To validate this grounding capacity beyond segmentation, we conduct additional *zero-shot* bounding-box localization experiments on two recent visual chain-of-thought style benchmarks: S-Chain (Le-Duc et al., 2025) and MedTrinity-25M (Xie et al., 2025). For S-Chain, we evaluate zero-shot performance on a randomly sampled subset of 100 images from the English test set. For MedTrinity-25M, since parts of the dataset are annotated using automated grounding models (as it is primarily designed for training), we restrict evaluation to a subset with *expert-annotated* bounding boxes and sample 100 images from this subset. Following the S-Chain protocol, spatial grounding is evaluated using bounding-box mIoU. Baselines include LISA (Lai et al., 2024), LISA++ (Yang et al., 2023), OMG-LLaVA (Zhang et al., 2024), and MedPLIB (Huang et al., 2025). The results are reported in Table 10.

*Table 10.* Zero-shot bounding-box grounding mIoU on visual chain-of-thought style benchmarks S-Chain and MedTrinity-25M.

| Model | mIoU (S-Chain) | mIoU (MedTrinity) |
|---|---|---|
| LISA | 8.8 | 7.7 |
| LISA++ | 9.6 | 7.3 |
| OMG-LLaVA | 12.1 | 7.6 |
| MedPLIB | 13.0 | 10.9 |
| MEDSIGHT | **14.0** | **16.2** |

MEDSIGHT achieves the best mIoU on both benchmarks, demonstrating that the proposed framework generalizes beyond segmentation to bounding-box grounding. This supports our claim that MEDSIGHT enables flexible spatial grounding within a unified generative framework, applicable to more complex visual chain-of-thought style spatial reasoning scenarios.

## F.6. Backbone Fairness with a 7B Variant

Our default MEDSIGHT configuration is built on the Qwen3-8B (Yang et al., 2025) LLM backbone, whereas many existing medical LVLM baselines use older or weaker 7B-scale models. To isolate the contribution of our architectural design from the choice of LLM backbone, we additionally train an alternative variant of MEDSIGHT built on Qwen2.5-7B [7], which aligns more closely with prior 7B-scale medical LVLMs. All other architectural components, training data, and training schedules are kept identical to the main MEDSIGHT configuration. We then compare both variants against representative baselines on VQA-RAD and DiagSeg in Table 11.

*Table 11.* Backbone fairness analysis. We provide a 7B variant of MEDSIGHT (Qwen2.5-7B) to align with the parameter scale of prior medical LVLMs, in addition to our default Qwen3-8B configuration.

| Model | VQA-RAD (close) | VQA-RAD (all) | DiagSeg-VQA | DiagSeg-Seg |
|---|---|---|---|---|
| LLaVA-Med | 60.2 | 48.1 | 35.3 | – |
| HuatuoGPT-Vision | 69.7 | 60.0 | 51.1 | – |
| OMG-LLaVA | 56.3 | 37.6 | 28.0 | 11.1 |
| MedPLIB | 58.3 | 34.8 | 13.1 | 31.8 |
| MEDSIGHT (Qwen2.5-7B) | 74.8 | 60.2 | **62.2** | 67.4 |
| MEDSIGHT (Qwen3-8B) | **79.9** | **61.4** | 58.9 | **69.9** |

---

[7]https://huggingface.co/Qwen/Qwen2.5-7B

Even when constrained to a 7B-scale backbone, MEDSIGHT (Qwen2.5-7B) consistently outperforms comparable medical LVLM baselines on both visual comprehension (VQA-RAD) and grounded diagnostic segmentation (DiagSeg). In particular, the 7B variant even surpasses the default Qwen3-8B variant on DiagSeg-VQA, indicating that the gains brought by our Region Perceiver and modality-aware codebook are not merely an artifact of using a stronger LLM. This confirms that the empirical improvements stem from the proposed design rather than from backbone scaling alone.

### F.7. Architectural Ablation of the Region Perceiver

The main ablation study in Table 5 compares the full MEDSIGHT model with a variant that completely removes the region embeddings $\mathbf{Q}_r$ from the LLM input. While this ablation directly demonstrates the importance of region tokens, it conflates two factors: (i) the architectural contribution of the Region Perceiver itself, and (ii) the additional segmentation supervision introduced by Region Perceiver pre-training. To disentangle these two factors, we further introduce a supervision-matched baseline that retains the same segmentation pretraining data and loss, but replaces the Region Perceiver with a simplified query-based decoder using *one-way* interaction. This simplified decoder removes the proposed dual cross-attention and the iterative refinement, while keeping the rest of the pipeline unchanged. The resulting simpler region features are used as a direct replacement for $\mathbf{Q}_r$ when feeding the LLM.

*Table 12.* Architectural ablation of the Region Perceiver. "Simpler Visual Features" shares the same segmentation supervision as MEDSIGHT but replaces the Region Perceiver's dual cross-attention and iterative refinement with a one-way query-based decoder.

| Setting | VQA-RAD (close) | VQA-RAD (all) | DiagSeg-VQA | DiagSeg-Seg |
|---|---|---|---|---|
| Removing $\mathbf{Q}_r$ | 72.8 | 59.4 | 54.4 | 59.2 |
| Simpler Visual Features | 76.0 | 59.7 | 56.6 | 63.8 |
| MEDSIGHT | **79.9** | **61.4** | **58.9** | **69.9** |

As shown in Table 12, the supervision-matched "Simpler Visual Features" baseline performs better than completely removing $\mathbf{Q}_r$, confirming that adding any form of segmentation supervision is beneficial. However, it still consistently underperforms the full MEDSIGHT model on both VQA-RAD and DiagSeg, indicating that the observed gains are *not* solely due to the extra segmentation supervision, but arise from the architectural design of the Region Perceiver itself.

### F.8. Codebook Size Ablation

The default MEDSIGHT model uses $M = 32$ codes per modality in the modality-aware region codebook. To understand the impact of the codebook size on both visual comprehension and grounding, we conduct an additional ablation in which we vary the number of codes per modality across $\{8, 16, 32, 64\}$ while keeping all other components fixed. We evaluate on VQA-RAD and DiagSeg, covering both visual comprehension and grounded diagnostic segmentation. Results are reported in Table 13.

*Table 13.* Effect of the codebook size per modality on visual comprehension (VQA-RAD) and grounded diagnostic segmentation (DiagSeg). The default setting used in the main paper is 32 codes per modality.

| Codebook size | VQA-RAD (close) | VQA-RAD (all) | DiagSeg-VQA | DiagSeg-Seg |
|---|---|---|---|---|
| 8 | 78.7 | 61.0 | 58.6 | 67.9 |
| 16 | 79.9 | 61.2 | 58.6 | 69.1 |
| 32 | 79.9 | 61.4 | **58.9** | 69.9 |
| 64 | 79.5 | 61.3 | 58.8 | **71.4** |

The codebook size has a clearly more pronounced impact on segmentation performance (DiagSeg-Seg) than on visual comprehension, which is consistent with our design motivation: the codebook primarily provides more expressive region tokens for spatial grounding. Smaller codebooks (e.g., 8 or 16 codes per modality) show degraded segmentation performance due to limited representational capacity. Larger codebooks (64 codes) provide marginal additional gains on segmentation but exhibit slight instability on VQA performance, suggesting diminishing returns and potential over-fragmentation. The chosen default of 32 codes per modality therefore offers a good trade-off between representational expressiveness and training stability, justifying the design choice used in the main paper.

