# OpenReview forum: "MedSIGHT: Towards Grounded Visual Comprehension in Medical Large Vision-Language Models"
_ICML.cc/2026/Conference — ICML 2026 regular_

### Official Review · Reviewer_gVYV · 2026-03-09

**Soundness:** 3
**Presentation:** 3
**Significance:** 3
**Originality:** 3
**Overall Recommendation:** 4
**Confidence:** 2

**Summary:**

This paper presents MedSIGHT, a medical large vision-language model capable of simultaneous vision–language comprehension and
medical image segmentation. The model processes medical images via a Region Perceiver to generate region-level embeddings; the patch-level and region-level embeddings are then mapped to a shared multimodal space through a projection layer, and combined with a large language model (LLM) augmented with a modality-aware Region Codebook for multimodal reasoning. The LLM simultaneously generates text-based diagnostic results containing region codes and pixel-level masks. It was trained on only 72K multimodal instruction pairs.  MedSIGHT has achieved state-of-the-art (SOTA) performance across multiple medical benchmarks and tasks involving various medical imaging modalities.

**Compliance With Llm Reviewing Policy:**

Affirmed.

**Final Justification:**

The author's rebuttal has resolved my concern. I will maintain my score and look forward to the author's camera-ready version, including their additional experiments.

**Key Questions For Authors:**

See above.

**Limitations:**

Yes

**Strengths And Weaknesses:**

Strengths:
1. The proposed method, MedSIGHT, got SOTA on various medical imaging modalities by only 72K multimodal instruction pairs.
2. The paper is well-organized and is clearly written.

Weaknesses:
1. Ablation experiments of Modality-aware Region Codebook size are missing.
2. If MedSIGHT is to be truly applied in hospitals or for disease diagnosis, I think it should be aligned with doctors' diagnoses. Could you provide a human expert alignment study with doctors on an OOD case?

---

> ### Author Rebuttal · Authors · 2026-03-30
>
> **\>\>\> W1:**
> We thank the reviewer for this valuable suggestion. We include an ablation study on the codebook size per modality to better understand its impact on performance. In the current model, we use 32 codes per modality. We extend this to evaluate multiple configurations: 8, 16, 64\. We conduct experiments on VQA-RAD and DiagSeg to evaluate visual comprehension and grounding ability.
>
> | Codebook size | VQA-RAD (close) | VQA-RAD (all) | DiagSeg-VQA | DiagSeg-Seg |
> | :------------ | :-------------- | :------------ | :---------- | :---------- |
> | 8             | 78.7            | 61.0          | 58.6        | 67.9        |
> | 16            | 79.9            | 61.2          | 58.6        | 69.1        |
> | 32            | 79.9            | 61.4          | 58.9        | 69.9        |
> | 64            | 79.5            | 61.3          | 58.8        | 71.4        |
>
> As shown in the results, the codebook size has a more pronounced impact on segmentation performance (**DiagSeg-Seg**), which aligns with our design motivation: the codebook primarily provides more expressive region tokens for segmentation. Specifically, smaller codebooks (e.g., 8, 16\) show a little degraded performance, especially on segmentation performance due to limited representational capacity. Larger codebooks (e.g., 64\) provide marginal gains but slight instability on VQA performance, suggesting diminishing returns and potential over-fragmentation. The current choice (32 codes per modality) achieves the best trade-off between expressiveness and stability.  These results will be included in the final version to justify the design choice of the modality-aware codebook.
>
> **\>\>\> W2:**
> We thank the reviewer for raising this important practical consideration. To evaluate alignment with expert reasoning in realistic and challenging settings, we conduct a qualitative study of 10 OOD clinical cases obtained from hospitals and not present in the training data. For each case, we prompt MedSIGHT to generate a detailed diagnostic reasoning process and result.
>
> A clinician is asked to evaluate the model’s diagnostic outputs by comparing them with the ground truth interpretations. The score range is from 1 to 5\. The higher, the better.  Across these cases, MedSIGHT achieves an average score of 3.8/5, indicating a good level of alignment with expert analysis in OOD cases. We will include detailed qualitative examples and case studies in the final version to further illustrate these findings.
>
> **We sincerely thank the reviewer again for the insightful feedback. We hope our responses have adequately addressed the concerns raised.**

---

> > ### Author Rebuttal · Reviewer_gVYV · 2026-04-03
> >
> > Thanks for the author's reply. I've decided to maintain my score.

---

> > > ### Author Response · Authors · 2026-04-03
> > >
> > > We are glad that we have addressed all the concerns raised in the review. We sincerely thank the reviewer for taking the time to carefully consider our rebuttal.

---

### Official Review · Reviewer_nRL3 · 2026-03-13

**Soundness:** 3
**Presentation:** 3
**Significance:** 2
**Originality:** 2
**Overall Recommendation:** 4
**Confidence:** 3

**Summary:**

MedSIGHT is a unified Med-LVLM that jointly performs visual comprehension and pixel-level segmentation within a single generative framework. The key architectural contributions are: a region perceiver that uses iterative dual cross-attention to produce spatially grounded region embeddings, enriching the LLM's visual input beyond standard patch-level CLIP features; a modality-aware region codebook that discretizes these embeddings via vector quantization and integrates them as new tokens in the LLM vocabulary, enabling the model to generate region-specific codes as part of its text output; and a four-stage progressive training pipeline that sequentially aligns each module before joint fine-tuning. The authors also introduce DiagSeg, a new benchmark that evaluates diagnosis-then-segmentation jointly, better reflecting real clinical workflows than existing benchmarks where segmentation targets are explicitly named in the prompt.

**Compliance With Llm Reviewing Policy:**

Affirmed.

**Final Justification:**

The rebuttal addressed all three of my concerns. My assessment remains a weak accept. The rebuttal was responsive and well-targeted, reinforcing rather than changing my prior evaluation.

**Key Questions For Authors:**

- Does a baseline exist where Qwen3-8B is fine-tuned on the same data but without the Region Perceiver or codebook?
- Could the authors report per-modality validation statistics or provide a second annotator's assessment, particularly given that DiagSeg spans eight modalities with very different diagnostic characteristics?
- Could the authors include a baseline that retains equivalent supervision but replaces Q_r with a simpler spatial feature, to isolate the architectural contribution of the region query mechanism?

**Limitations:**

Yes

**Strengths And Weaknesses:**

Strengths

- The dual cross-attention in the region perceive, where image features are updated by region queries and vice versa, is a well-motivated architectural design.
- The modality-aware codebook design is a practical and sensible response to the heterogeneity of medical imaging that is absent from general-domain analogues.
- Treating generated region codes as vocabulary tokens whose LLM hidden states decode into masks eliminates the need for an external segmentation module, resulting in a cleaner end-to-end framework.
- DiagSeg is a meaningful benchmark contribution. The diagnosis-then-segmentation framing better reflects real clinical reasoning than prompt-specified segmentation tasks.

Weaknesses

- MedSIGHT uses Qwen3-8B while most baselines use older or weaker 7B models. The paper does compare against Qwen3-VL (same family), but that model is not fine-tuned on medical data, so the comparison still conflates architecture with domain adaptation. No ablation fixes the backbone and removes the Region Perceiver or codebook to isolate their contribution to comprehension performance.
- DiagSeg benchmark validation is insufficient. Human validation covers only 100 of 1,655 samples (~6%) assessed by a single physician with no inter-rater agreement reported. For a benchmark proposed for community reuse, this is a thin reliability guarantee.
- Supervision confound in region token ablation: The ablation removing Q_r shows performance drops, but does not disentangle whether the gain comes from the region query mechanism itself or from the additional segmentation supervision that the Region Perceiver pretraining introduces. A fairer baseline would retain equivalent segmentation supervision while replacing Q_r with a simpler spatial feature, to isolate the architectural contribution.

---

> ### Author Rebuttal · Authors · 2026-03-30
>
> **\>\>\> W1 & Q1:**
> We thank the reviewer for raising concerns about backbone fairness and the ablation study.
>
> **Backbone control.** We include an additional version of MedSIGHT built on Qwen2.5-7B, which aligns more closely with prior 7B-scale medical MLLMs. The results in the following table show that MedSIGHT (Qwen2.5-7B) consistently outperforms comparable baselines, indicating that the improvements are not solely due to the stronger Qwen3-8B backbone.
>
> | Model                 | VQA-RAD (close) | VQA-RAD (all) | DiagSeg-VQA | DiagSeg-Seg |
> | :-------------------- | :-------------- | :------------ | :---------- | :---------- |
> | LLaVA-Med             | 60.2            | 48.1          | 35.3        | –           |
> | HuatuoGPT-Vision      | 69.7            | 60.0          | 51.1        | –           |
> | OMG-LLaVA             | 56.3            | 37.6          | 28.0        | 11.1        |
> | MedPLIB               | 58.3            | 34.8          | 13.1        | 31.8        |
> | MedSIGHT (Qwen2.5-7B) | 74.8            | 60.2          | 62.2        | 67.4        |
> | MedSIGHT (Qwen3-8B)   | 79.9            | 61.4          | 58.9        | 69.9        |
>
> **Ablation with fixed backbone.** We’d like to point out that we have such ablation study results in Table 5 on Page 8 (lines 389-391), where we fixed backbone and removed either Region Perceiver $\mathbf{Q}_r​$ or Codebook $\mathcal{C}$. The results show that removing either component leads to substantial performance degradation.
>
> **\>\>\> W2 & Q2:**
> To strengthen the validation of DiagSeg, following the reviewer’s suggestion, we include a second medical expert for independent annotation. Besides, we increase the number of validated samples, ensure balanced coverage across all modalities by sampling 30 cases per modality (240 cases in total). Finally, we report inter-rater agreement using Pearson and Spearman correlations to measure the agreement between annotators. The modality-level validation results are as follows:
>
> | Modality     | Expert A Mean | Expert B Mean | Pearson r | Spearman r |
> | :----------- | ------------: | ------------: | --------: | ---------: |
> | CT           |          4.90 |          5.00 |       n/a |        n/a |
> | Dermatoscopy |          4.13 |          4.20 |     0.869 |      0.896 |
> | Endoscopy    |          5.00 |          5.00 |       n/a |        n/a |
> | MRI          |          4.80 |          4.67 |     0.775 |      0.777 |
> | OCT          |          4.97 |          5.00 |       n/a |        n/a |
> | Pathology    |          4.83 |          4.90 |     0.745 |      0.745 |
> | Ultrasound   |          5.00 |          5.00 |       n/a |        n/a |
> | X-Ray        |          4.50 |          4.53 |     0.935 |      0.935 |
>
> Note that **n/a** means correlation is undefined because one or both raters have zero variance in that modality (all scores identical). Across all 240 samples, Expert A and Expert B achieve mean scores of 4.77 and 4.79, respectively. The higher Pearson and Spearman correlations show strong agreement among annotators across eight modalities, supporting the benchmark's reliability. We will include those results in the final version.
>
> **\>\>\> W3 & Q3:**
>
> We thank the reviewer for this insightful comment. To validate the effect of the architecture design in Region Perceiver, as suggested by the reviewer, we introduce a new ablation study baseline that isolates the architectural contribution.
>
> Specifically, we retain the same segmentation pretraining data and loss and replace the Region Perceiver with a simplified query-based decoder using one-way interaction, removing the proposed dual cross-attention and iterative refinement. The resulting simpler region features are used as a direct **replacement** for $\mathbf{Q}_r​$ and the rest of the pipeline remains unchanged.
>
> | Setting                 | VQA-RAD (close) | VQA-RAD (all) | DiagSeg-VQA | DiagSeg-Seg |
> | :---------------------- | :-------------- | :------------ | :---------- | :---------- |
> | Removing $\mathbf{Q}_r​$           | 72.8            | 59.4          | 54.4        | 59.2        |
> | Simpler Visual Features | 76.0            | 59.7          | 56.6        | 63.8        |
> | MedSIGHT                | 79.9            | 61.4          | 58.9        | 69.9        |
>
> We also use “Removing $\mathbf{Q}_r$” as a baseline to demonstrate the necessity of the Region Perceiver design. The Simpler Visual Features baseline, while stronger than the ablation setting that removes $\mathbf{Q}_r​$,  performs consistently worse than MedSIGHT on both VQA-RAD and DiagSeg benchmarks. This indicates that the observed gains are not solely due to segmentation supervision, but arise from the proposed Region Perceiver design. We will include these results in the final version to provide a clearer analysis of architectural contributions.
>
> **We sincerely thank the reviewer for the constructive feedback. We hope our responses have adequately addressed the concerns raised.**

---

> > ### Author Rebuttal · Reviewer_nRL3 · 2026-04-03
> >
> > All three of my concerns have been adequately addressed by the rebuttal. In light of these responses, I will like to maintain my positive score.

---

> > > ### Author Response · Authors · 2026-04-03
> > >
> > > We are very glad that our responses have addressed your concerns. We sincerely thank the reviewer for carefully reviewing our rebuttal and for the continued positive score.

---

### Official Review · Reviewer_9Wj7 · 2026-03-13

**Soundness:** 3
**Presentation:** 3
**Significance:** 3
**Originality:** 3
**Overall Recommendation:** 4
**Confidence:** 5

**Summary:**

The paper addresses the problem of **(i) insufficient visual detail** in which Med-LVLMs rely on CLIP patch features that capture high-level semantics but lose fine spatial details needed for accurate medical understanding; and **(ii) limited output representation**: Using a single token for all segmentation regions restricts the model’s ability to distinguish different anatomical or pathological areas.

The first point (i) is addressed by the author by proposing a Region Perceiver, grouping patch-level tokens into group-level token and (ii) is learned by mapping continous tokens (mixing between patch-level and Region Perceiver tokens) into a set of discrete token sets via Codebook, which are later represented as new vocabulary for the LLM model. The author validates model performance on visual perception tasks (VQA) and grounded diagnostic segmentation, comparing performance with general/medical MLLMs as well as unified multi-modal models having seperate segmentation models.

**Compliance With Llm Reviewing Policy:**

Affirmed.

**Key Questions For Authors:**

I would increase my rating if the authors could support additional results of MedSight on the two weakness points mentioned above.

**Limitations:**

Please consider adding a limitation section to discuss the current weaknesses of the MedSight.

**Strengths And Weaknesses:**

**A. Strengths**

**i. Motivation & Presentation:**  The paper is clearly written and well organized. The motivation is easy to follow, and the authors provide intuitive explanations for the design choices (e.g., limitations of CLIP patch features and single segmentation tokens). The architecture and training pipeline are also well illustrated, making the proposed framework accessible to readers.

**ii. Method novelties:** The ideas of quantizing continous tokens into a set of discrete ones and treating them as a set of new vocabulary tokens learned in pre-training LLM are very exciting in my opinion. Though related concepts, such as region queries, perception, or discrete visual tokens, exist in the broader vision-language community; however, integrating region tokens as part of the LLM vocabulary for generative grounded reasoning in medical LVLMs is relatively novel in the medical domain.

Furthermore, the author also contributes a new curated dataset toward a triplet sample (image, output text, and segmentation masks).

**iii. High Records on Zero-shot Performance:** The paper demonstrates strong zero-shot generalization across medical VQA and segmentation tasks, achieving competitive results with relatively limited instruction data. This suggests the proposed representation learning improves cross-task and cross-modality generalization.

**B. Weakness**

The main concern of the Reviewer is about the experiments to fully convince them of the method's effectiveness.

**a. Missing evaluation under realistic downstream adaptation settings.**
 All experiments in the paper focus on zero-shot inference, and the manuscript **lacks experiments evaluating fine-tuning scenarios**, which are common in real-world medical applications due to the domain gap between pretraining data and downstream clinical datasets. The strong zero-shot performance only evaluates the representation quality, not the final system performance. Therefore, it would be valuable to include experiments with full fine-tuning or parameter-efficient methods (e.g., LoRA adapters). To avoid the unfairness of model size, where several baselines have very large parameter counts, the author can restrict comparisons to recent medical MLLMs of similar scale (e.g., ~7B–8B parameters), such as models in [1], thereby further strengthening the benchmark.

**b. Limited evaluation of spatial grounding beyond segmentation.**

**Most experiments focus on segmentation outputs, while in clinical workflows, bounding-box localization is often a more practical and challenging task**, especially when lesions visually resemble nearby tissues and require precise localization for physician verification. Although the paper states that the framework supports both segmentation and bounding boxes, no experiments validate this claim. Therefore, it would also be interesting to evaluate the model under visual chain-of-thought grounding settings (e.g., S-Chain dataset [2] or MedTrinity-25M [3]) to demonstrate the framework’s ability to support more complex spatial reasoning scenarios. The author can sample a subset of this dataset if time is limited, demonstrate how MedSight works on it, and compare it with a few principal models, which should be enough.

This experiment would significantly strengthen the paper's contributions both in the methodology aspect (doing well so far) and in practical impacts.

[1] ExGra-Med: Extended Context Graph Alignment for Medical Vision-Language Models, NeurIPS 2025

[2] S-Chain: Structured Visual Chain-of-Thought for Medicine, 2025

[3] MedTrinity-25M: A Large-scale Multimodal Dataset with Multigranular Annotations for Medicine, ICLR 2025

---

> ### Author Rebuttal · Authors · 2026-03-30
>
> **\>\>\> W1:**
> We thank the reviewer for this important suggestion. Following the reviewer’s recommendation, we have conducted additional experiments under parameter-efficient fine-tuning (LoRA) settings on three widely used medical VQA benchmarks: SLAKE, VQA-RAD, and PathVQA, using the same settings as in ExGra-Med \[1\]. To ensure fairness, we restrict comparisons to recent medical MLLMs of comparable scale, including LLaVA-Med 1.5, HuatuoGPT-Vision 7B, and ExGra-Med. Specifically, we apply LoRA to all linear layers with rank 32 for all models. **All models are fine-tuned under the same training setting and data splits**. For metrics, we use Accuracy for close-ended VQA and Recall for open-ended VQA. The detailed results are as follows:
>
>
> | Model               | SLAKE (close) | SLAKE (open) | VQA-RAD (close) | VQA-RAD (open) | PathVQA (close) | PathVQA (open) |
> | :------------------ | :------------ | :----------- | :-------------- | :------------- | :-------------- | :------------- |
> | LLaVA-Med 1.5       | 88.5          | 83.3         | 74.4            | 36.7           | 93.2            | 38.0           |
> | HuatuoGPT-Vision 7B | 90.1          | 85.6         | 76.8            | 41.2           | 93.3            | 37.1           |
> | ExGra-Med           | 88.9          | 85.1         | 75.2            | 38.9           | 93.3            | 37.9           |
> | MedSIGHT (ours)     | 93.2          | 89.9         | 82.3            | 43.9           | 94.2            | 38.1           |
>
> As shown in the results, MedSIGHT consistently outperforms all baselines across the three benchmarks. This demonstrates that the proposed architecture not only provides strong zero-shot representations, but also maintains superior performance under downstream adaptation. We will include these results in the final version to provide a more comprehensive evaluation.
>
> **\>\>\> W2:**
>
> We thank the reviewer for highlighting the necessity of evaluating spatial rounding on bounding-box localization. We would like to clarify that MedSIGHT, like unified grounding models such as OMG-LLaVA and MedPLIB, supports bounding-box localization by **deriving bounding boxes from the decoded segmentation masks**. This is a standard practice in grounding-based frameworks.
>
> To further validate this grounding capacity, following the reviewer’s suggestion, we conduct additional **zero-shot experiments** as follows. For S-Chain, we evaluate zero-shot performance on a randomly sampled subset of 100 images from the English test set. For MedTrinity-25M, since parts of the dataset are annotated using automated grounding models (as it is primarily designed for training), we restrict evaluation to a subset with **expert-annotated** bounding boxes, sampling 100 images. The baselines include LISA, LISA++, OMG-LLaVA, and MedPLIB. Following the S-Chain protocol, spatial grounding is evaluated using bounding-box **mIoU**.
>
> | Model     | mIoU (S-Chain) | mIoU (MedTrinity) |
> | :-------- | :------------- | :---------------- |
> | LISA      | 8.8            | 7.7               |
> | LISA++    | 9.6            | 7.3               |
> | OMG-LLaVA | 12.1           | 7.6               |
> | MedPLIB   | 13.0           | 10.9              |
> | MedSIGHT  | 14.0           | 16.2              |
>
> As shown in the results, MedSIGHT demonstrates better performance than grounding baselines, indicating that the proposed framework generalizes beyond segmentation to bounding-box grounding. This supports our claim that MedSIGHT enables flexible spatial grounding within a unified generative framework. We will include these results in the final version to further strengthen the empirical validation of the method.
>
> **We sincerely thank the reviewer again for the constructive feedback. We hope our responses have sufficiently addressed the concerns raised.**

---

> > ### Author Rebuttal · Reviewer_9Wj7 · 2026-04-03
> >
> > Thank you, authors, for your efforts to address my concerns. Everything is resolved, and I will keep my current positive evaluations.

---

> > > ### Author Response · Authors · 2026-04-03
> > >
> > > We are glad that our responses have addressed your concerns. We sincerely thank the reviewer for carefully reviewing our rebuttal and for the continued positive consideration.

---

### Decision · Program_Chairs · 2026-04-30

**Decision:**

Accept (regular)

**Comment:**

All reviewers found the proposed method to be novel and the results promising. The rebuttals fully resolved the reviewers' remaining concerns. The AC read the reviews and rebuttals, and concur with the reviewers' assessment.